# Circadian clock disruption promotes the degeneration of dopaminergic neurons in male *Drosophila*

Michaëla Majcin Dorcikova ®[1], Lou C. Duret[1], Emma Pottié[1] & Emi Nagoshi ®[1] ✉

Sleep and circadian rhythm disruptions are frequent comorbidities of Parkinson's disease (PD), a disorder characterized by the progressive loss of dopaminergic (DA) neurons in the substantia nigra. However, the causal role of circadian clocks in the degenerative process remains uncertain. We demonstrated here that circadian clocks regulate the rhythmicity and magnitude of the vulnerability of DA neurons to oxidative stress in male *Drosophila*. Circadian pacemaker neurons are presynaptic to a subset of DA neurons and rhythmically modulate their susceptibility to degeneration. The arrhythmic *period* (*per*) gene null mutation exacerbates the age-dependent loss of DA neurons and, in combination with brief oxidative stress, causes premature animal death. These findings suggest that circadian clock disruption promotes dopaminergic neurodegeneration.

Disturbances in the sleep-wake rhythm are well-described symptoms in neurodegenerative disorders such as Parkinson's disease (PD)[1]. PD is a complex disorder characterized by the progressive loss of dopaminergic (DA) neurons in the substantia nigra pars compacta, leading to motor symptoms[2]. Sleep and circadian disruption can be observed years before the onset of motor symptoms[3], raising the possibility that circadian rhythm disruption precedes and contributes to PD development. Studies using animal models of PD showed that circadian disruption could exacerbate neurodegeneration and motor symptoms[4]. However, it remains elusive whether circadian disturbances causally drive neurodegeneration or are caused by neurodegeneration, partly because PD pathology includes dysfunction of the brain regions regulating sleep, resulting in disrupted sleep-wake cycles[5]. Dissecting mechanistic links between circadian clock disruption and neurodegeneration—two intertwined and potentially synergistic processes—is critical for developing reliable diagnostic tools and treatment strategies for PD.

*Drosophila melanogaster* is a genetically tractable model system for studying neurodegenerative diseases[6]. Taking advantage of the conservation of PD-related molecular genetic pathways, numerous genetic and toxin-induced models of PD that display varying degrees of neurodegeneration and motor and non-motor symptoms have been generated[7,8]. The adult fly midbrain contains approximately eight

clusters of DA neurons projecting to various brain regions[9]. Functional impairments or loss of DA neurons in the protocerebral anterior medial (PAM) and protocerebral anterior lateral 1 (PPL1) clusters have been consistently observed in PD models, in addition to the losses in other clusters[10–15]. The PAM cluster comprises ~130 neurons per hemisphere, accounting for approximately 80% of fly brain DA neurons. PAM neurons are highly heterogeneous in their projection patterns and functions and play critical roles in behavioral processes such as olfactory associative learning, foraging, sleep, and locomotion[16–20]. We and others showed that the loss of a subclass of PAM neurons leads to defective startle-induced climbing behavior in several PD models[13,14,21], suggesting a partial analogy between the PAM cluster and the substantia nigra pars compacta.

Circadian clocks, built upon the conserved design principle of negative feedback loops, drive rhythms in numerous behavioral and physiological processes in species across phylogenic trees. The core feedback loop of the *Drosophila* circadian clock consists of CLOCK/CYCLE (CLK/CYC) heterodimers that activate transcription of the *period* (*per*) and *timeless* (*tim*) genes and PER and TIM proteins that feedback-inhibit CLK/CYC activity. CLK/CYC also activates the transcription of genes encoding VRILLE (VRI) and PDP-1, negatively and positively regulating *Clk*, forming a stabilizing loop interlocked with the core loop[22]. Circadian clocks are present in approximately 150

[1]Department of Genetics and Evolution and Institute of Genetics and Genomics of Geneva (iGE3), University of Geneva, CH-1211 Geneva, Switzerland. ✉e-mail: Emi.Nagoshi@unige.ch

pacemaker neurons and 1800 glia in the fly brain[23]. Pacemaker neurons are classified into functionally and anatomically diverse subgroups: small and large lateral ventral neurons (s- and l-LNvs), lateral dorsal neurons (LNds), lateral posterior neurons, and three groups of dorsal neurons (DN1s, DN2s, and DN3s). The s- and l-LNvs express the neuropeptide pigment-dispersing factor (PDF), essential for synchronizing rhythmicity across the pacemaker circuit. The PDF-positive s-LNvs (the M-cells) drive the morning peak of activity under light-dark cycles (LD) and control free-running 24-h period locomotor rhythms in constant darkness (DD)[24,25]. The robustness and flexibility of the locomotor behavior adapting to environmental conditions require the communication between M-cells and the second set of pacemaker subtypes, the E-cells, comprising the PDF-negative fifth s-LNv, half of the LNds, and some of the DN1s (reviewed in ref. [26]). The pacemaker circuit controls circadian locomotor output and regulates brain-wide rhythmic physiology by conveying time-of-day information to different areas in the brain, such as the mushroom bodies (MBs)[27] and the posterior medial protocerebrum 3 (PPM3) cluster DA neurons[28,29].

Misalignment between environmental cycles and endogenous circadian rhythms increases the risk for physical and psychiatric disorders[30]. Genetic and environmental disruption of circadian rhythms in flies leads to adverse health consequences, including increased mortality in response to oxidative stress[31] and lifespan reduction[32], suggesting that the mechanisms underpinning the link between circadian disruption and diseases can be studied in flies. *Clk* loss of function in the s-LNv pacemaker neurons causes loss of DA neurons in the PPL1 cluster, resulting in age-dependent locomotor decline; intriguingly, this process is PDF-dependent and circadian-clock-independent[33]. Therefore, questions remain regarding whether and how the circadian system regulates the (patho-)physiology of DA neurons.

Here, we used *Drosophila* as a model to ask whether circadian clocks causally impact degenerative processes in PD. We found that DA neurons in the PAM cluster exhibit diurnal and circadian rhythms in vulnerability to oxidative stress. Loss of circadian clock genes results in age-dependent PAM neuron loss and exacerbates oxidative stress-induced PAM neuron loss. We identified PAM-$\alpha$1 neurons as one of the susceptible DA neuron subtypes and found that the circadian pacemaker circuit is presynaptic to this subgroup. Oxidative stress-induced loss of PAM-$\alpha$1 neurons leads to changes in sleep, reminiscent of PD non-motor symptoms. These findings establish a direct role of circadian clocks in controlling the timing and magnitude of the vulnerability of DA neurons.

## Results
### Circadian vulnerability of DA neurons to oxidative insults
We previously demonstrated that flies exposed to hydrogen peroxide ($H_2O_2$) or paraquat for 24 h display selective loss of DA neurons in the PAM cluster (Fig. 1a)[12,13,21]. To evaluate dopaminergic neurodegeneration from a circadian perspective, we developed a short-term $H_2O_2$ treatment protocol that can be performed at six time points across the day. Seven-day-old $w^{1118}$ flies were treated with $H_2O_2$ at various concentrations (5–20%) and durations (3–6 h), and survival of PAM neurons was assessed 7 days post-treatment by immunostaining for tyrosine hydroxylase (TH) (Supplementary Fig. S1a). From the conditions that induced PAM neuron loss, we chose 5% and 10% $H_2O_2$ treatment for 4 h in subsequent experiments because this duration is suitable for performing assays in a circadian fashion, and the effect does not saturate. Loss of TH-positive neurons was observed only in the PAM cluster among all DA neuron clusters following the short $H_2O_2$ treatment (Fig. 1b). $H_2O_2$ treatment in flies expressing red nuclear fluorescent protein RedStinger with the PAM neuron-specific *R58E02* GAL4 driver resulted in the reduction of RedStinger-positive cells, suggesting that oxidative insults cause loss of neurons and not merely the reduction in TH expression

(Supplementary Fig. S1b). Therefore, we focused on this group of DA neurons in subsequent studies.

To test whether the vulnerability of PAM neurons to oxidative stress is rhythmic, we next performed the short $H_2O_2$ treatment in a circadian fashion. Seven-day-old $w^{1118}$ flies entrained to LD (12 h:12 h light-dark cycles) were exposed to 10% $H_2O_2$ for 4 h, following a 5-h food and water deprivation. $H_2O_2$ exposure was initiated at a different Zeitgeber Time (ZT) across the day, and DA neurons were analyzed 7 days post-treatment (Fig. 1c). Whereas the $H_2O_2$ treatment at any time caused significant PAM neuron loss compared to the water-only control, the treatment at ZT20 (8 h after lights-off) resulted in a more significant loss of PAM neurons than the other time points (Fig. 1d). The highest sensitivity at ZT20 was replicated in the assay with the 4-h 5% $H_2O_2$ treatment, validating its specificity (Supplementary Fig. S1c). We performed the same experiment in constant darkness (DD) to determine the role of endogenous clocks and LD cycles. Newly hatched flies were entrained for 3 days in LD and then placed in DD for the remainder of the experiment and treated with $H_2O_2$ on the fourth day in DD (DD4) (Fig. 1c). The treatment at CT4 (4 h after subjective lights-on) with 10% (Supplementary Fig. 1c) or 5% $H_2O_2$ (Supplementary Fig. S1d) caused a more significant PAM neuron loss than the other time points. These results suggest intrinsic rhythmicity in the vulnerability of PAM neurons to oxidative stress modulated by light.

To determine whether circadian clocks control the intrinsic rhythms of PAM neuron vulnerability, we next used *per* null (*per^01^*, also known as *per^0^*) mutant flies carrying *white* mutation, devoid of molecular clocks. We first quantified PAM neuron counts in *per^0^* with anti-TH staining and by expressing RedStinger without oxidative stress from days 1 to 14 of age. $w^{1118}$ was used as a control strain. We found that *per^0^* flies were born with fewer PAM neurons and displayed a marked, age-dependent loss of PAM neurons (Fig. 2a–c). The *tim* null (*tim^0^*) mutants, also in $w$ background, similarly displayed a reduced number of PAM neurons on day 1 and a progressive loss thereafter (Supplementary Fig. S2a). Both *per^0^* and *tim^0^* arrhythmic mutants displayed locomotor deficits, as reported previously[33,34] (Supplementary Fig. S2b). These findings suggest that disruption of circadian clocks interferes with the development of DA neurons in the PAM cluster and accelerates their age-dependent degeneration.

Because a previous study reported that $H_2O_2$ exposure elicits a higher rate of mortality in *per^0^* than in wild-type flies[34], we also examined the effect of various concentrations of $H_2O_2$ on PAM neuron survival in *per^0^* flies at a single timepoint; 2.5% $H_2O_2$, which did not affect PAM neurons in $w^{1118}$ and *Canton-S* (*CS*), robustly induced PAM neurodegeneration in *per^0^* mutants (Fig. 2d). This finding suggests that loss of *per* null mutation enhances the vulnerability of PAM neurons to oxidative stress-induced degeneration. We next performed the circadian oxidative treatment with 5% $H_2O_2$ in LD and DD on *per^0^* flies. The *per^0^* flies treated with $H_2O_2$ at any timepoint showed a significant loss of PAM neurons as compared to the control group treated with water only. In LD, the treatment at ZT12 resulted in a more significant loss of PAM neurons than the other time points (Fig. 2e). In contrast, the difference among time points was abolished in DD (Fig. 2f). These findings suggest that the circadian clock gene *per* controls the magnitude and temporal variations of PAM neuron sensitivity to oxidative stress. In the absence of functional clocks, light can impose diurnal rhythms of PAM neuron vulnerability.

### The circadian neural circuit controls the rhythmic vulnerability of PAM neurons
Because PAM neurons are not clock-containing cells, the rhythmic vulnerability of PAM neurons is non-cell-autonomously controlled by clocks located elsewhere. Several mechanisms might drive rhythms in non-clock-containing neurons, including the rhythmic $H_2O_2$ uptake by

feeding/drinking rhythms, systemic rhythmic changes in the brain environment, and the rhythmic modulation of the physiology of PAM neurons by circadian pacemaker neurons.

To test whether the differential loss of PAM neurons is caused by ingesting more $H_2O_2$ solution at one time than the others, we performed a feeding assay[35]. Flies were fed with $H_2O_2$ or water mixed with blue food dye, which sticks to the gut and is not eliminated or digested. Then, spectrophotometry was performed on homogenized fly bodies to measure the amount of dye ingested. In LD and DD, flies displayed rhythmic feeding patterns when fed with the control solution (water

and dye), congruent with a previous study demonstrating the circadian regulation of feeding[36]. In contrast, the feeding pattern of the 10% $H_2O_2$-containing solution was not rhythmic. Its uptake was significantly lower than the control solution across all time points (Fig. 3a). This finding is consistent with the finding that 2.5% $H_2O_2$ caused a more significant neuronal loss than the higher concentration in $per^0$ flies (Fig. 2d), probably because flies ingest more $H_2O_2$ solutions at lower concentrations. Therefore, daily variations in the feeding pattern do not account for the circadian vulnerability of PAM neurons. We also found that flies drink three to five times more in DD than in LD, which

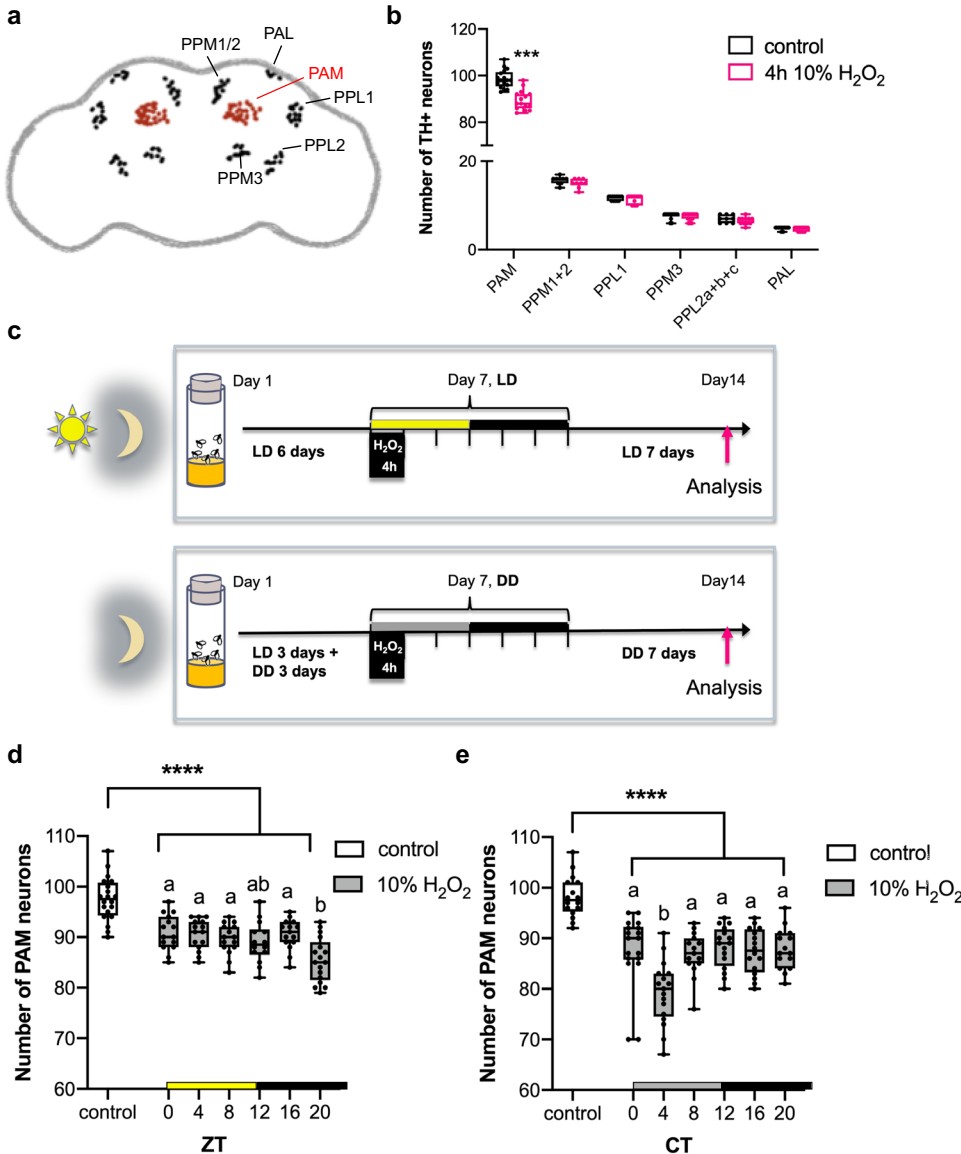

**Fig. 1 | Rhythmic vulnerability of PAM neurons to oxidative stress. a** A schematic of the cell bodies of DA neuron clusters in the adult fly brain. PAM protocerebral anterior medial, PAL protocerebral anterior lateral, PPL1 and 2 protocerebral posterior lateral 1 and 2, PPM1, 2, and 3 protocerebral posterior medial 1, 2, and 3. **b** DA neuron counts in $w^{1118}$ flies per hemisphere in each cluster 7 days after a 4-h 10% $H_2O_2$ treatment performed at ZT20. The control group was treated with water only. DA neurons were detected by anti-TH immunostaining. Neurodegeneration was observed only in the PAM cluster following the $H_2O_2$ treatment ($n = 18$ hemispheres analyzed for both groups). ***$p < 0.0001$ (two-tailed $t$-test, comparing the control and $H_2O_2$-treated groups). In box plots in this and all following figures, box boundaries are the 25th and 75th percentiles, the horizontal line across the box is the median, and the whiskers indicate the minimum and maximum values. The dots represent all data points. **c** A schematic of the circadian

$H_2O_2$ treatment protocol. **d, e** Quantification of the number of PAM neurons after the 4-h 10% $H_2O_2$ treatment performed at different times in LD (control, $n = 20$; ZT0, 4, 8 and 16 $n = 15$ each; ZT12, $n = 14$; ZT16, $n = 17$ hemispheres) (**d**) or DD (control, $n = 16$; CT0, $n = 18$, CT4, $n = 17$; CT8 and 20, $n = 15$ each; CT12 and 16, $n = 16$ each) (**e**). The $x$-axis indicates the times when $H_2O_2$ was applied. The control group was treated with water only at ZT20 in LD (**d**) and CT20 in DD (**e**). At all time points, PAM neuron counts in the $H_2O_2$ treatment group are significantly smaller than those in the control group. ****$p < 0.0001$ (one-way ANOVA with Tukey's post hoc test). Within the $H_2O_2$-treated group, flies treated at ZT20 in LD (**d**) and CT4 in DD (**e**) showed significantly greater cell loss than the treatment at any other time. Different lowercase letters represent statistical significance by one-way ANOVA with Tukey's HSD post hoc test. Source data are provided as a Source Data file.

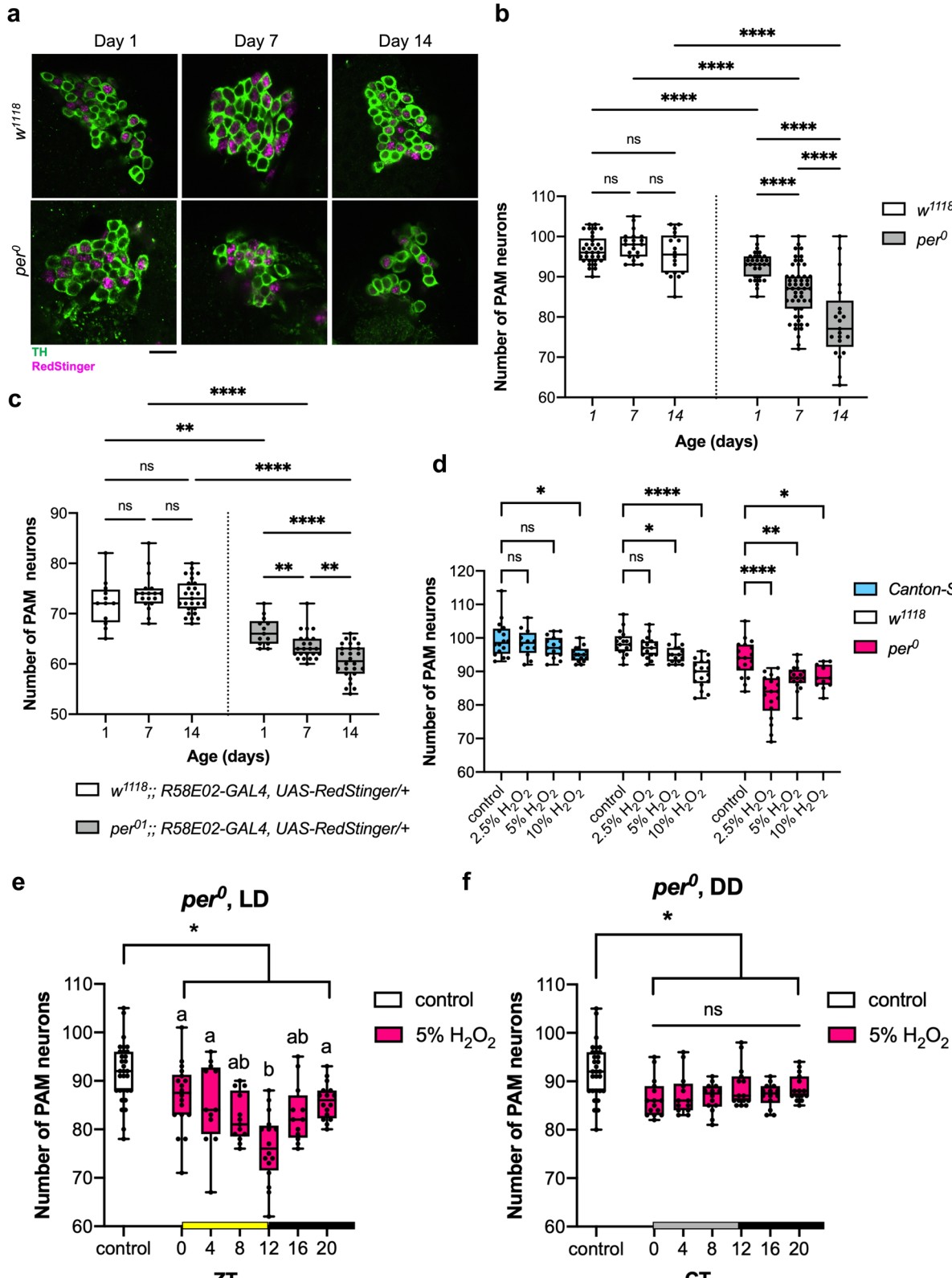

explains the more pronounced loss of PAM neurons following the $H_2O_2$ treatment in DD than in LD (Fig. 1d, e, Supplementary Fig. S1c, d).

The circadian $H_2O_2$ feeding assay requires flies to be starved before and during $H_2O_2$ exposure. We next tested whether the starvation and dehydration associated with the assay led to the rhythmic PAM neurodegeneration. To this end, flies were removed from food and water for 9 h, 5 h before and 4 h after ZT20. Seven

days later, PAM neurons were examined by anti-TH immune staining. The control group was given access to water for 4 h after a 5-h starvation/water deprivation period. The results showed no changes in PAM neuron count between the groups (Fig. 3b). This finding suggests that the loss of PAM neurons and their rhythmicity are not caused by starvation or dehydration before or during the $H_2O_2$ treatment.

**Fig. 2 | The clock gene *per* controls the magnitude and rhythms of the vulnerability of PAM neurons to oxidative stress. a** Representative maximum-projection images of PAM neurons in $w^{1118}$ and $per^0$ flies at indicated ages. PAM neurons were visualized by anti-TH antibodies (green) and RedStinger driven by the *R58E02* driver (magenta). Approximately half of the neurons in the PAM cluster are visible. Scale bar, 20 μm. **b, c** The number of PAM neurons detected by anti-TH immunostaining ($w^{1118}$ day1, $n = 33$; day7, $n = 21$; day14, $n = 18$ hemispheres. $per^0$ day1, $n = 31$; day7, $n = 52$; day14, $n = 21$.) (**b**) and by the expression of *R58E02*-driven RedStinger ($w^{1118}$ day1, $n = 12$; day7, $n = 17$; day14, $n = 27$ hemispheres. $per^0$ day1, $n = 13$; day7, $n = 23$; day14, $n = 26$.) (**c**). $per^0$ flies display developmental and age-dependent loss of PAM neurons. **$p < 0.01$ and ****$p < 0.0001$ (two-tailed *t*-test or two-tailed Mann–Whitney U test). **d** PAM neuron counts in *Canton-S*, $w^{1118}$, and $per^0$ were assessed by anti-TH staining 7 days after the treatment with different concentrations of $H_2O_2$. Treatment was performed from ZT1 for 4 h. (*Canton-S* control, $n = 16$, 2.5% $H_2O_2$, $n = 12$; 5%, $n = 15$; 10%, $n = 16$ hemispheres. $w^{1118}$ control, $n = 17$; 2.5%, $n = 16$; 5%, $n = 16$; 10%, $n = 17$. $per^0$ control, $n = 18$; 2.5%, $n = 18$; 5%, $n = 17$; 10%,

$n = 11$.) $per^0$ flies display increased PAM neuron susceptibility compared to control genotypes. *$p < 0.05$, ***$p < 0.001$, and ****$p < 0.0001$ (one-way ANOVA with Dunnett's multiple comparisons test). **e, f** PAM neuron counts in $per^0$ flies after a 4-h 5% $H_2O_2$ treatment were performed at different time points in LD (control, $n = 31$; ZT0, $n = 18$, ZT4, $n = 12$; ZT8, $n = 13$; ZT12, $n = 16$; ZT16, $n = 12$; ZT20, $n = 16$ hemispheres) (**e**) or DD (control, $n = 30$; CT0, $n = 15$; CT4, $n = 13$; CT8, $n = 14$; CT12, $n = 15$; CT16, $n = 14$; CT20, $n = 15$ hemispheres) (**f**). The *x*-axis indicates the time points when $H_2O_2$ was applied. The control group was treated with water only at ZT0 in LD (**e**) and CT0 in DD (**f**). At all time points, PAM neuron counts in the $H_2O_2$ treatment group were significantly smaller than those in the control group. *$p < 0.05$ (one-way ANOVA with Tukey's HSD post hoc test). Within the $H_2O_2$-treated group, flies treated at ZT12 in LD (**e**) displayed a significantly greater cell loss than at any other time point. No difference was observed between time points in DD (**f**). Different lowercase letters represent statistical significance by one-way ANOVA with Tukey's post hoc test. Source data are provided as a Source Data file.

The results obtained so far suggest that the loss of PAM neurons can be directly attributed to the increased levels of reactive oxygen species (ROS) in PAM neurons resulting from $H_2O_2$ treatment rather than the systemic effects triggered by $H_2O_2$ ingestion. To test this further, we expressed catalase, an enzyme that breaks down H2O2 into water and oxygen, specifically in the PAM neurons using the *R58E02* driver. The expression of catalase effectively prevented the $H_2O_2$-induced loss of PAM neurons in both LD (treated at ZT20) and DD (at CT4 and CT16) (Fig. 3c, d). These results indicate that the elevation of ROS within the PAM neurons directly leads to the neurodegeneration of PAM neurons following exposure to $H_2O_2$.

Endogenous oxidative stress levels vary daily in flies and mammals[31,37]. We wondered whether the rhythms in endogenous ROS levels in the fly brain and $H_2O_2$ administered at specific times would accumulate to trigger a rhythmic neuronal loss in the PAM cluster. To test this possibility, we used MitoSOX (a ROS-sensitive fluorescent dye targeted to mitochondria) to monitor mitochondrial ROS levels across the day in PAM neurons labeled with *R58E02>UAS-EGFP* in $w^{1118}$ flies. In 7-day-old flies, MitoSOX fluorescence intensity in PAM neurons in LD showed daily variations and peaked at ZT20 (Fig. 3e), the time when PAM neurons are the most vulnerable to $H_2O_2$ (Fig. 1d). This finding suggests that oscillating levels of endogenous ROS combined with the $H_2O_2$ treatment might account for the differential loss of PAM neurons in LD. In contrast, no significant differences in MitoSOX levels were observed between time points in DD (Fig. 3e, Supplementary Fig. S3). Therefore, another mechanism under the control of circadian clocks gates the vulnerability of PAM neurons in DD.

To test whether PER-expressing clock neurons directly or indirectly modulate PAM neuron vulnerability, we re-expressed the PER protein in clock neurons in $per^0$ mutants and examined PAM neurons. Genetic rescue of *per* with the pan-clock neuron driver *Clk1982-GAL4*[38] significantly increased the number of PAM neurons at day 14 compared to $per^0$ without $H_2O_2$ treatment. Thus, the pan clock-neuron *per* rescue prevented the accelerated age-dependent loss of PAM neurons caused by loss of *per* (Fig. 4a). *Clk1982>per* also reversed the exacerbated loss of PAM neurons following $H_2O_2$ treatment in $per^0$ to the level of $w^{1118}$ flies (Fig. 4b). These results suggest that loss of PER in clock neurons is responsible for the enhanced PAM neuron loss in $per^0$ mutants, at least in part.

To test if any subclass of clock neurons is essential for the role of PER in regulating PAM neuron survival, we next performed $per^0$ genetic rescue using *Pdf-GAL4*, expressed in the s- and l-LNvs except the fifth s-LNv[39,40], and *DvPdf-GAL4*, which targets all LNvs, three CRY-negative LNds, and one CRY/ITP double-positive LNd[41,42]. Both rescue genotypes restored the number of PAM neurons to the level of $w^{1118}$ under basal conditions (Fig. 4a). The number of surviving PAM neurons after $H_2O_2$ treatment was also significantly greater in both rescue genotypes than in control expressing only the driver (Fig. 4b). PER rescue with

*Pdf-GAL4* or *Clk1982-GAL4* showed similar levels of protection against $H_2O_2$, whereas, with *DvPdf-GAL4*, the protection was partial. This might be due to differential expression levels within the LNvs between *Pdf-GAL4* and *DvPdf-GAL4*.

Additionally, we examined whether the genetic rescue of $per^0$ in clock neurons could prevent PAM neuron loss in DD. *Clk1982-GAL4* is highly expressed in most clock neurons, except in the LPN, but has an ectopic expression in non-clock cells, such as the Kenyon cells. Therefore, we included a second pan-clock neuron driver, *Clk856-GAL4*, which covers LPN and has a reduced ectopic expression[38] in this experiment. $H_2O_2$ treatment was performed at CT4 and CT16. The number of PAM neurons was significantly higher in all the rescue genotypes compared to $per^0$ at both time points (****$p < 0.0001$ for each pairwise comparison between $per^0$ and the rescue genotypes). Flies rescued with either *Clk1982-GAL4* or *DvPdf-GAL4* exhibited partial protection against $H_2O_2$-induced PAM neuron loss, whereas flies rescued with *Clk856-GAL4* or *Pdf-GAL4* were completely resistant to $H_2O_2$ at both time points (Fig. 4c). These findings suggest that *Clk856-GAL4* and *Pdf-GAL4* may drive the expression of PER above endogenous levels, thereby bolstering the resistance of PAM neurons beyond that in wild-type flies. Taken together, these findings suggest that PER expression within the PDF-positive LNvs plays a significant role in the regulation of survival of PAM neurons following oxidative insults.

### A subset of clock neurons are presynaptic to PAM neurons

The finding that PER expressed in circadian pacemaker neurons controls rhythmic PAM neuron vulnerability suggests that PAM neurons are directly or indirectly downstream of circadian neural circuits. To test this possibility, we examined postsynaptic targets of *DvPdf-GAL4*-expressing neurons using *trans*-Tango[43], an anterograde transsynaptic tracing tool. Several neurons in the most anterior region of the PAM cluster were marked with the postsynaptic signal, indicating the direct projection from the *DvPdf-GAL4*-labeled neurons to 6 to 7 PAM neurons (Fig. 4d). In contrast, the *trans*-Tango assay with *Pdf-GAL4* did not result in any positive signal within the PAM cluster (Fig. 4d), suggesting that the s- and l-LNv neurons are not presynaptic to PAM neurons. These findings taken together suggest that all or some *DvPdf*-positive but *Pdf*-negative cells (i.e., the three CRY-negative LNds, one CRY/ITP double-positive LNd, and the fifth LNv) are presynaptic to PAM neurons (Fig. 4e).

### Identification of the vulnerable subpopulation of PAM neurons

PAM neurons are highly heterogeneous, encompassing 14 subclasses of morphologically and functionally diverse neurons projecting to different subdomains of the MB and other brain regions[44]. Because the number of neurons lost following $H_2O_2$ treatment was relatively consistent between experiments within the same genotype, we wondered if the vulnerable neurons belong to a specific subset(s) of PAM

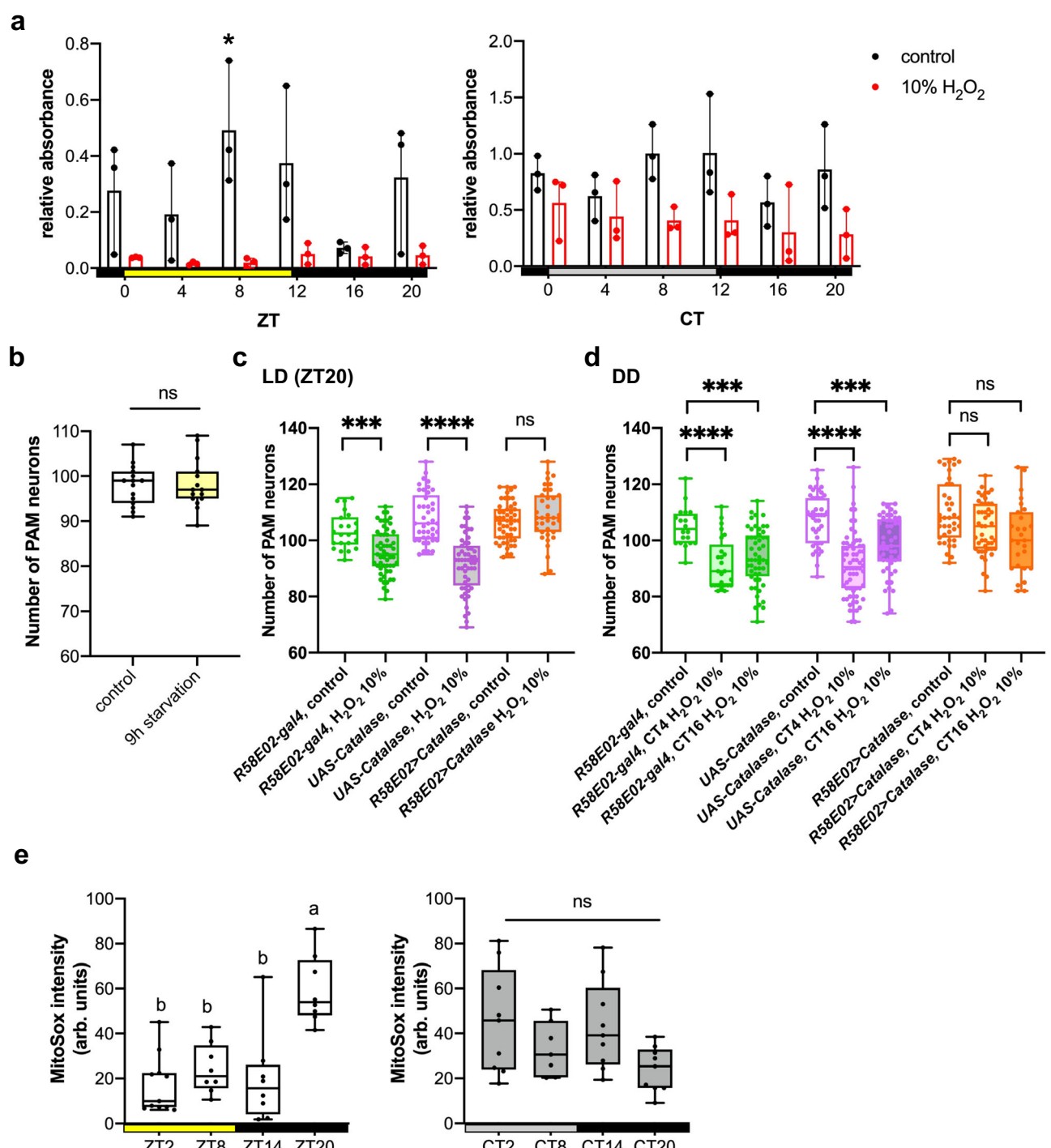

neurons. To test this possibility, we first used classical GAL4 drivers to label most or some PAM neurons with GFP to roughly locate the subclass vulnerable to $H_2O_2$ administered at ZT20. Most (8 of 13) PAM neurons that degenerated after the $H_2O_2$ insult belonged to the group expressing *R58EO2-GAL4* (Fig. 5a).

To identify the vulnerable PAM subgroup within the *R58EO2*-positive cells, we tested split-GAL4s. From the available split-GAL4 drivers created at the Janelia Research Campus[45], we selected all lines with activation domains driven with the *R58EO2* promoter, i.e., MB312B, MB043B, MB299B, and MB315C. These drivers label the neurons innervating different but partly overlapping areas of the MBs: MB312B-neurons send projections to the γ-lobe (γ4 subdomain), MB043B-neurons to the α- β-, and β'-lobes (α1, β1, β'1ap, and β'1m

subdomains), MB299B neurons to the α-lobe (α1 subdomain) and MB315C-neurons to the γ-lobe (γ5 subdomain)[44]. *UAS-GFP* was expressed with each of these drivers, and the number of remaining GFP-positive cells after short-term $H_2O_2$ treatment was scored. Neuronal loss was observed within the clusters labeled by the MB315C and MB299B drivers (Fig. 5b, c). We tested one more split-GAL4 driver (MB441B) that does not contain the *R58EO2*-driven component but is expressed in the PAM neuron subset projecting to the MB γ-lobe (γ3 subdomain). $H_2O_2$ treatment caused a loss of neurons within the subpopulation labeled by MB441B (Fig. 5b, c). These results suggest that PAM neurons innervating the MB γ3, γ5, and α1 subdomains (PAM γ3, γ5, and α1 subgroups, respectively) are subpopulations vulnerable to neurodegeneration after oxidative insults (Fig. 5d). Of these, the

**Fig. 3 | Oxidative stress elicited by H$_2$O$_2$ ingestion but neither feeding rhythms nor dehydration causes circadian vulnerability of PAM neurons. a** Feeding patterns of $w^{1118}$ flies in LD (left) and DD (right). Flies were fed for 4 h with 10% H$_2$O$_2$ solution containing blue dye. The control solution contained only water and blue food dye. $n = 15$–25 flies. The $x$-axis indicates the time points when the flies started to be fed. The $y$-axis represents the relative absorbance per ten flies. The dots represent the values of three independent experiments. Feeding levels in the control group at ZT8 were significantly higher than at other time points in LD. *$p < 0.05$ in any pairwise comparison (ANOVA with Tukey's post hoc test). Error bars indicate the range. No significant difference was observed between time points in all other groups. **b** PAM neuron counts after 9 h of food and water deprivation, started at ZT15. The control group was deprived of food and water for 5 h, followed by 4 h of water access. $n = 15$ hemispheres for both groups. No significant difference between groups ($t$-test). **c, d** Catalase was expressed in PAM neurons with the $R58E02$ driver, and its effect on H$_2$O$_2$-induced PAM neuron loss was assessed by anti-TH immunostaining in LD (**c**) and DD (**d**). Catalase expression prevented PAM neuron loss following a 4-h 10% H$_2$O$_2$ treatment performed at ZT20 (**c**), as well as at CT4 and CT16 (**d**). The control group was treated with water at the same time point. ***$p < 0.001$ ($p = 0.0004$ in (**c**); $p = 0.0004$ between $R58E02$ control and CT16 H$_2$O$_2$, $p = 0.0006$ $UAS$-$Catalase$ control vs. CT16 H$_2$O$_2$ in (**d**)) and ****$p < 0.0001$ (Kruskal–Wallis test with Dunn's multiple comparisons test). In (**c**), $R58E02$ control, $n = 22$; H$_2$O$_2$, $n = 54$ hemispheres. $UAS$-$Catalase$ control, $n = 41$; H$_2$O$_2$, $n = 55$. $R58E02 > Catalase$, $n = 50$; H$_2$O$_2$, $n = 37$. In (**d**), $R58E02$ control, $n = 20$; CT4 H$_2$O$_2$, $n = 25$; CT16 H$_2$O$_2$ $n = 52$ hemispheres. $UAS$-$Catalase$ control, $n = 43$; CT4 H$_2$O$_2$, $n = 68$; CT16 H$_2$O$_2$, $n = 49$. $R58E02 > Catalase$ control, $n = 39$; CT4 H$_2$O$_2$, $n = 47$; CT16 H$_2$O$_2$, $n = 27$. **e** ROS levels within the PAM neurons were measured using MitoSOX red in flies expressing $EGFP$ with the $R58E02$ driver in LD (left) and DD (right). ZT2, $n = 11$; ZT 8, 14 and 20; 8 flies. CT2, 14 and 20, $n = 9$; CT8, $n = 7$. ROS levels are significantly elevated at ZT20 in LD but do not differ among time points in DD (ANOVA with Tukey's post hoc test). Source data are provided as a Source Data file.

PAM α1 subgroup labeled by MB299B was the most vulnerable, as more than 50% degenerated after the H$_2$O$_2$ treatment (Fig. 5b–d).

We next determined whether these PAM subpopulations were rhythmically vulnerable to oxidative stress, focusing on the most vulnerable MB299B-labeled neurons (PAM-α1). We treated 7-day-old flies expressing GFP with the MB299B split-GAL4 at ZT8 and ZT20 with 10% H$_2$O$_2$, and the GFP-positive cell count was scored 7 days later. The H$_2$O$_2$ treatment at ZT8 caused the loss of around three to four neurons, whereas, on average, 7 neurons were lost from the treatment at ZT20. Thus, PAM-α1 indeed shows rhythmicity in vulnerability to oxidative stress, at least in LD (Fig. 5e, f). In DD, exposure to H$_2$O$_2$ at CT4 and CT16 both induced a significant loss of the MB299B-labeled neurons, but no difference between time points was observed (Fig. 5g). This finding suggests the presence of another PAM subpopulation that displays rhythmic vulnerability in DD.

We also analyzed the Janelia Research Campus hemibrain connectome data[46], which were visualized using the NeuPrint interface[47] to assess the structural connectivity between clock neurons and MB299B neurons. Although chemical synapses between the LNvs or LNds and PAM-α1 neurons are not annotated in the hemibrain data set[46], visual inspection using the NeuPrint tool revealed close contacts between the axonal projections of the LNds and MB299B neurons (Fig. 6a), whereas projections of the s-LNvs and MB299B neurons did not show apparent contacts (Supplementary Fig. S4a). These findings are consistent with the results of the *trans*-Tango experiment (Fig. 4d) and suggest that input from the LNds modulates the vulnerability of MB299B neurons in a circadian fashion. We found that the projections of the LNds also contacted MB315C and MB441 neurons, supporting the notion that vulnerable PAM neuron subsets are postsynaptic to the LNds (Supplementary Fig. S4b, c).

### Rhythmically vulnerable PAM subpopulations exhibit calcium rhythms

The finding that MB299B-positive, PAM-α1 neurons are rhythmically vulnerable to oxidative insults in LD and postsynaptic to LNds raises the possibility that they exhibit physiological rhythms. Therefore, we measured levels of intracellular Ca$^{2+}$, a second messenger that controls several cellular functions, in PAM-α1 neurons across 24 h in live flies (Fig. 6b). The calcium sensor GCaMP7s[48] was expressed with the MB299B driver, and the 3-day-old flies were entrained to LD cycles for four days before imaging in DD. Ca$^{2+}$ levels in these neurons showed temporal variations, with peaks twice a day, around CT5 and CT17 (Fig. 6c). In $per^0$ flies, GCaMP fluorescence levels were uneven across time points, and the bimodal pattern in $w^{1118}$ was lost, with a peak around CT17 (Fig. 6d). These results suggest that circadian inputs to the PAM-α1 neurons rhythmically modulate their activity that peaks just before midday and before midnight.

We next wanted to know whether decreased excitability could be neuroprotective. Therefore, we silenced PAM neurons by overexpressing the inward rectifier K+ channel, Kir2.1[49], with the $R58E02$ driver and tested whether this silencing prevents the degeneration of PAM neurons after the oxidative stress challenge. However, significantly fewer PAM neurons were present in the 14-day-old flies expressing Kir2.1 than in the control without H$_2$O$_2$ treatment, probably due to developmental impairments. Moreover, Kir2.1 overexpression did not prevent H$_2$O$_2$-induced PAM neuron loss (Supplementary Fig. S5). This finding suggests that neuronal activity levels are unlikely to be the cause of the selective and rhythmic vulnerability of PAM-α1 neurons.

### Consequences of PAM-α1 neurodegeneration on motor and non-motor functions

Four-hour treatment with 10% H$_2$O$_2$ at ZT20 leads to the loss of approximately 13 DA neurons in the PAM cluster, most of which belong to the PAM-α1 subclass (Fig. 5a–d). We wondered whether this loss of approximately 10% of PAM neurons would have immediate or long-term effects on motor or non-motor functions. It was reported that synaptic dysfunction of a small subset of PAM neurons (expressing $NP6510$-$GAL4$) could impair the fly's ability to climb[14]. To test whether the PAM-α1 subgroup has a similar role in climbing ability, we performed a startle-induced negative geotaxis assay[21] after the short-term H$_2$O$_2$ treatment with increasing age. Climbing abilities of control and treated flies declined with age, as expected and as previously reported[14]. However, there was no difference between the control and experimental groups up to day 56, i.e., 49 days post-H$_2$O$_2$ treatment (Supplementary Fig. S6), except for day 49. These findings suggest that PAM-α1 is not involved in the control of climbing behavior.

DA neurons play various roles in regulating behavior and physiology, including sleep. Dopaminergic input to the MB regulates several aspects of sleep[50]. Because the role of the MB α1 compartment on sleep remains elusive, we next examined the effect of H$_2$O$_2$-induced loss of PAM-α1 on sleep. Following the short-term H$_2$O$_2$ treatment on day 7 in LD, flies were maintained under LD cycles, and sleep was analyzed over 3 days starting at days 11 and 17. The results showed an increase in total sleep duration in H$_2$O$_2$-treated flies in both age groups (Fig. 7a, b). Whereas total sleep duration was increased both during the day and the night, mean sleep episode duration was increased after the H$_2$O$_2$ treatment only during the night, indicating that consolidation of sleep occurred specifically during the night. This effect was more pronounced in older flies (Fig. 7c, d). In contrast, activity levels of the flies during the waking period were not different between the H$_2$O$_2$-treated and non-treated groups at both ages (Fig. 7e, f). These findings indicate that a 4-h H$_2$O$_2$ treatment increases sleep but not hypoactivity.

H$_2$O$_2$ administration may affect sleep independently of its effect on PAM-α1 neurons. To distinguish these possibilities, we measured sleep while blocking the output of PAM-α1 neurons using a temperature-sensitive mutant of dynamin, $Shibire^{ts}$ ($Shi^{ts}$). $UAS$-$Shi^{ts}$

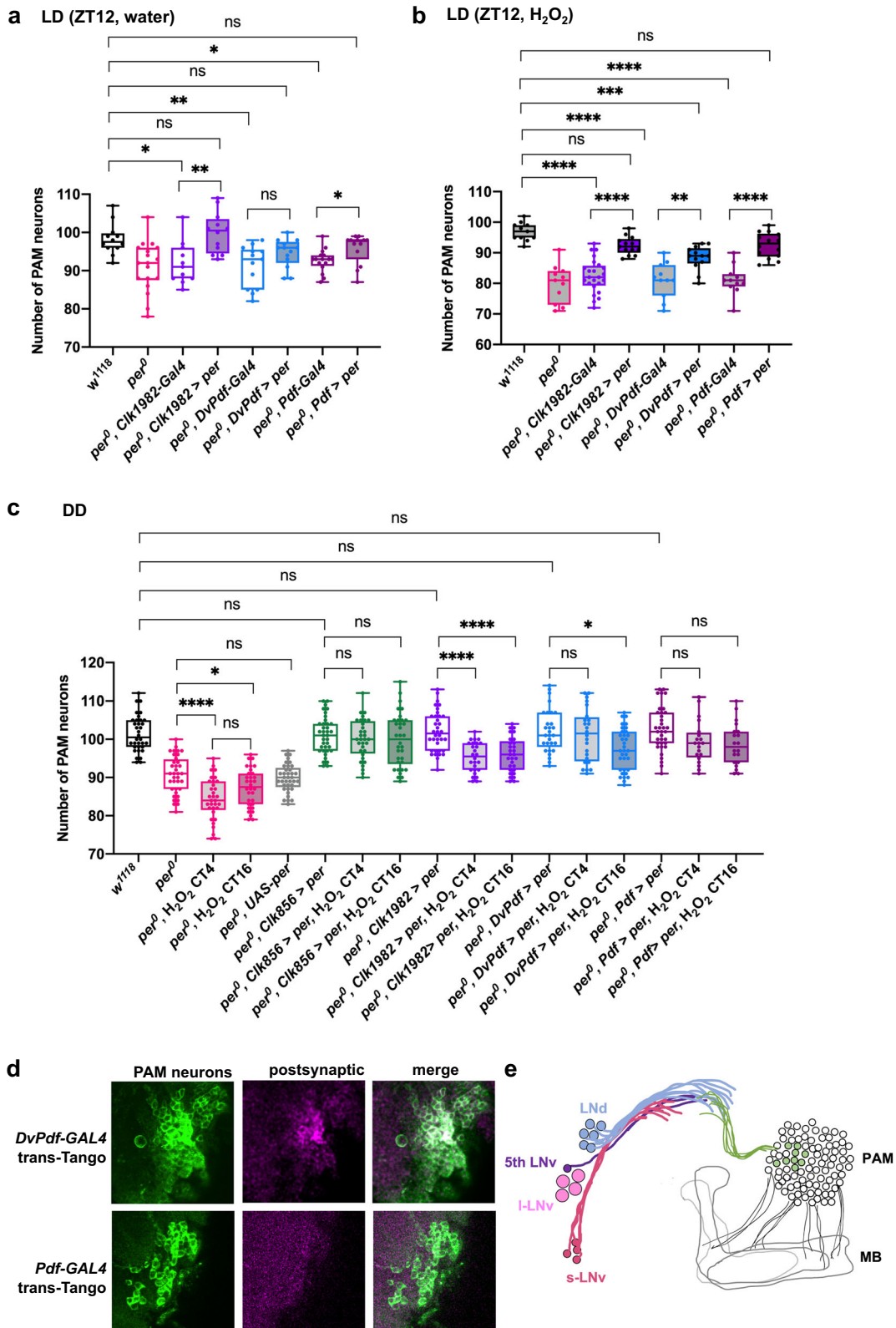

**d** PAM neurons, postsynaptic, merge

*DvPdf-GAL4* trans-Tango

*Pdf-GAL4* trans-Tango

**e** LNd, 5th LNv, l-LNv, s-LNv, PAM, MB

was driven with MB299B, and sleep and activity were monitored under 30 °C between day 11 and 13 after eclosion in flies that were raised at 19 °C. In comparison to control flies carrying only *UAS-Shi^ts^*, *MB299B>Shi^ts^* flies showed an increase and decrease in total sleep duration during the day and the night, respectively (Supplementary Fig. S7a). Sleep was more consolidated during the day (Supplementary Fig. S7b); however, the mean activity count during the waking period

was reduced across 24 h, indicating hypoactivity (Supplementary Fig. S7c). Thus, inhibition of PAM- α1 synaptic output not only reduces activity levels but also modulates sleep. Although the effect on sleep by H₂O₂ exposure and synaptic block are not identical, possibly due to the temperature difference and the incomplete blockage of neuropeptide release by *Shi^ts51,52^*, these results collectively suggest the involvement of PAM-α1 neurons in the regulation of sleep and activity. However, it

**Fig. 4 | PER expression in clock neurons preserves PAM neurons. a–c** Genetic rescue of *per* in clock neurons in *per⁰* mutants. PAM neurons were counted using anti-TH staining 7 days after a control treatment with water (**a**) and after a 4-h treatment with 2.5% $H_2O_2$ performed at ZT12 in LD (**b**). In (**c**), the control group was treated with water at CT16, and 2.5% $H_2O_2$ treatment was performed at CT4 and CT16. Genotypes are indicated on the *x*-axis. a > b represents that UAS-transgene b is driven by the GAL4 driver a. *n* = 11–24 hemispheres (see Source Data for individual sample numbers). Rescue genotypes significantly improve the preservation of PAM neurons after the $H_2O_2$ treatment. *$p < 0.05$, **$p < 0.01$, ***$p < 0.001$, and ****$p < 0.0001$ (one-way ANOVA with Dunnett's multiple comparisons test, or Kruskal–Wallis test with Dunn's multiple comparisons test). All the rescue genotypes in (**c**) exhibit a significantly higher number of PAM neurons compared to *per⁰*

(****$p < 0.0001$ for each pairwise comparison, not shown in the figure for clarity). *n* = 11–19 (**a**), *n* = 11–24 (**b**), and *n* = 20–39 hemispheres (**c**). **d** *trans*-Tango experiments using *DvPdf-GAL4* and *Pdf-GAL4* combined with TH immunostaining (green). Several postsynaptic targets of *DvPdf* neurons (magenta) were identified as PAM neurons, whereas no postsynaptic signal of *Pdf* neurons was found within the PAM cluster. Presynaptic signals by *DvPdf-GAL4* and *Pdf-GAL* are not visible in these pictures. Two independent experiments. Scale bar, 20 μm. **e** A schematic representation of the *trans*-Tango labeling results. Projections of the l-LNvs are not shown for clarity. *DvPdf-GAL4*- positive but *Pdf-GAL4*-negative neurons, i.e., LNds, are presynaptic to a subset of PAM neurons. Source data are provided as a Source Data file.

remains to be determined if the effects on sleep by the $H_2O_2$ treatment result from the loss of PAM- α1 neurons.

## The implication of the multiple-hit hypothesis for Parkinson's disease

PD arises from the interaction of genetic and environmental risk factors and age. This notion, formulated as the multiple-hit hypothesis[53], suggests that short-term $H_2O_2$ treatment is merely a single "hit" to the dopaminergic system. To test this idea, we quantified the number of PAM neurons in the $H_2O_2$-treated flies over the course of aging. The difference between the control and treated groups was significant 7 days after $H_2O_2$ treatment and was maintained at least up to day 56 (49 days post-treatment), with no acceleration of neurodegeneration in either of the two groups (Fig. 8a).

These findings support the multiple-hit hypothesis and suggest that an insult in addition to $H_2O_2$ might accelerate PAM neurodegeneration. We speculated that *per⁰* mutation might constitute the second "hit," following our finding that *per⁰* mutants display age-dependent PAM neurodegeneration (Fig. 2a–c). Therefore, we counted the number of PAM neurons over the course of aging in *per⁰* flies after the $H_2O_2$ treatment. On day 14, *per⁰* showed a substantial loss of PAM neurons as expected; however, after that, the rate of PAM neurodegeneration in the $H_2O_2$-treated flies was not higher than in non-treated flies, reaching the same number of PAM neurons on day 28 (Fig. 8b).

Increased mortality is a characteristic of PD patients[54]. We monitored the life spans of *w¹¹¹⁸* and *per⁰* flies with and without $H_2O_2$ treatment. Whereas the $H_2O_2$ treatment did not affect the life spans of *w¹¹¹⁸* flies, $H_2O_2$-treated *per0* flies exhibited a significantly reduced lifespan compared to the control group (Fig. 8c, d). These results demonstrate that the combination of an oxidative insult and the *per* null mutation causes premature death of the animal.

## Discussion

Circadian rhythm disruptions are frequent comorbidities in neurodegenerative disorders; nevertheless, little is known about how circadian clocks and neurodegeneration are causally related. Here we showed that the circadian clock gene *per* controls daily rhythms and the magnitude of the vulnerability of DA neurons to oxidative insults in *Drosophila*. The circadian pacemaker circuit is presynaptic to the vulnerable subpopulation of DA neurons, PAM-α1, and rhythmically modulates its intracellular $Ca^{2+}$ levels. Oxidative stress-induced dopaminergic neurodegeneration leads to persistent changes in sleep and, combined with the *per* null mutation, causes premature death of the animal. Taken together with the finding that *per* null mutation drives progressive loss of DA neurons, our results suggest a causal role of circadian clocks in dopaminergic neurodegeneration.

PAM-α1 neurons are highly vulnerable and rhythmically susceptible to oxidative stress. What makes this subpopulation more susceptible than other DA neurons and how circadian inputs gate the vulnerability remain questions for future studies. The LNd clock neurons are presynaptic to PAM-α1 neurons, and the latter exhibit *per*-dependent intracellular $Ca^{2+}$ rhythms. Because $Ca^{2+}$ levels in neuronal

cell bodies can be correlated with firing frequency[55], one can speculate that neuronal activity rhythms gated by the pacemaker circuit temporally regulate the vulnerability of PAM-α1. However, constant hyperpolarization by the expression of the Kir2.1 channel did not prevent $H_2O_2$-induced PAM neuron loss. The LNds express the neuropeptide ion transport peptide (ITP), the short neuropeptide F (sNPF), and acetylcholine (Ach)[56]. The neuromodulatory input by the inhibitory sNPF[57] or ITP, rather than the excitatory Ach, likely regulates the vulnerability of PAM neurons, given that neuronal silencing does not prevent PAM neuron loss. Identifying molecular differences between PAM-α1 and other subpopulations and potential molecular rhythmicity within PAM-α1 will be necessary to understand the mechanisms underlying these neurons' selective and rhythmic vulnerability. Of note, PAM-α1 neurons exhibit rhythmic susceptibility to $H_2O_2$ in LD, while no rhythms in vulnerability were observed in DD. These findings suggest the presence of another subgroup of PAM neurons that exhibit vulnerability rhythms peaking at CT4. Future studies aimed at characterizing rhythms in distinct PAM subpopulations will provide further insights into this phenomenon.

$Ca^{2+}$ rhythms in MB299B-labeled neurons are bimodal, peaking around CT5 and CT17. This observation can be interpreted in two ways: all MB299B neurons exhibit bimodal $Ca^{2+}$ rhythms, or there are two populations of neurons within the MB299B-labeled neurons that are anti-phasic to one another. The former possibility is reminiscent of a previous finding that the M- and E- cells of the circadian circuit[29] drive bimodal $Ca^{2+}$ rhythms in the PPM3 cluster of DA neurons. The latter possibility gains support from the fact that, although PAM-α1 is the primary subtype labeled with the MB299B driver, weak expression of MB299B was also detected in PAM- *β1* and *β2* neurons[17,58]. Curiously, $Ca^{2+}$ levels in the MB299B neurons in *per⁰* show temporal variation with a peak around CT17, although its pattern is irregular and dissimilar to that of the *w¹¹¹⁸* flies. What causes this variation is unknown, but the CT17 peak might be an after-effect of the LD entrainment. Although further studies are required to untangle these issues, our observation is nevertheless consistent with the notion of connectivity between the circadian system and PAM neurons. Interestingly, the connectome data and results of the *trans*-Tango experiments suggest that the LNds are directly connected to the MB299B-labeled neurons but not the s-LNvs. However, restoring *per* expression within the PDF-positive LNvs is sufficient to reverse the PAM neuron loss in the basal conditions and under oxidative stress in *per⁰* mutants. Because PDF-positive s-LNvs can synchronize the LNds through PDF/PDFR signaling[59], these results suggest that the presence of PER within the PDF-positive s-LNvs is sufficient to convey a protective signal to the MB299B neurons via LNds.

Arrhythmic *per⁰* and *tim⁰* mutants display age-dependent loss of PAM neurons. Although both mutants completely disrupt molecular clocks, the observed phenotype could be a gene-specific effect rather than the result of loss of circadian clocks per se. A previous study demonstrated that *Clk* deficiency within the s-LNvs causes age-dependent loss of DA neurons in the PPL1 cluster and accelerates locomotor decline; however, the other clock gene loss-of-function

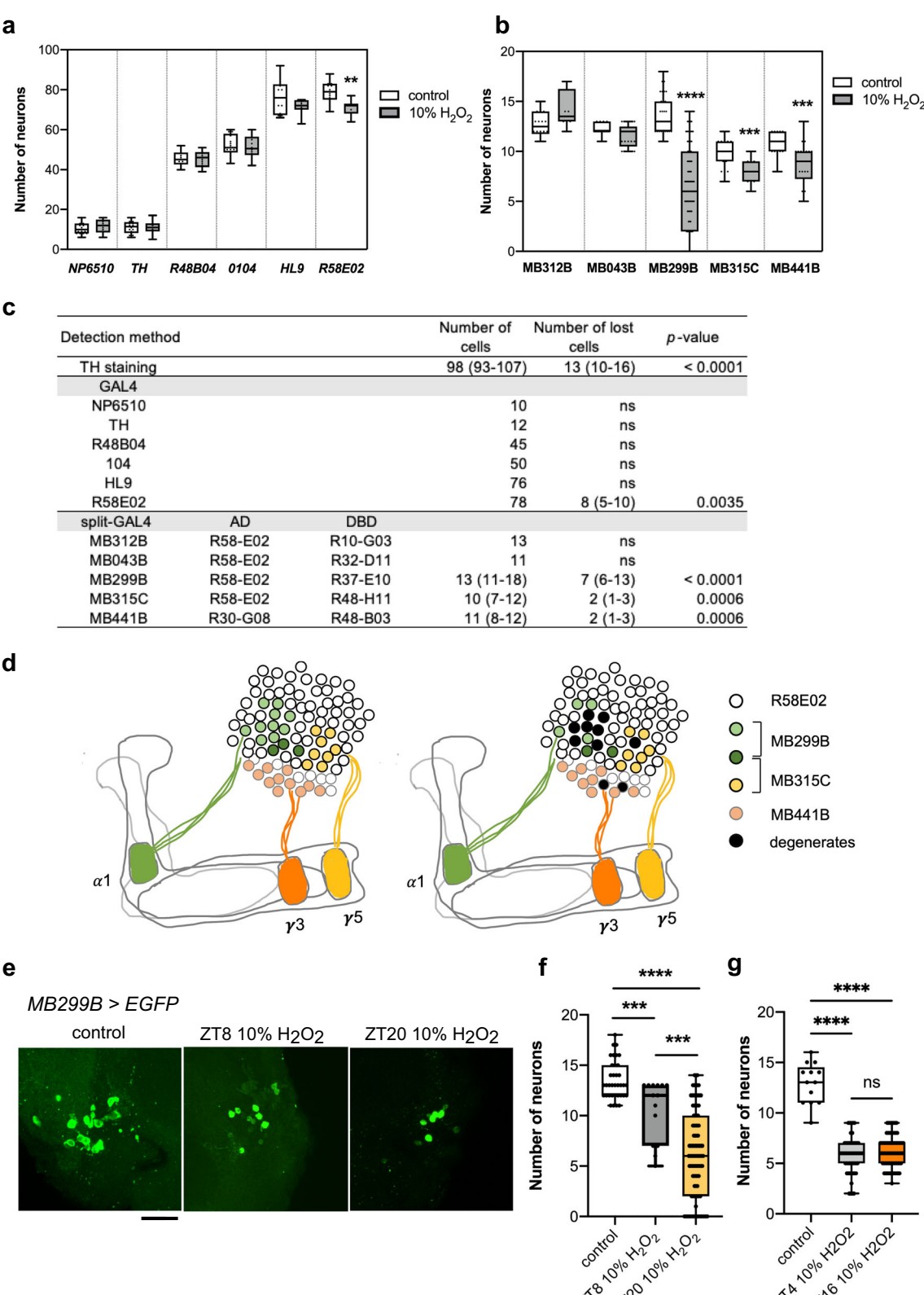

c

| Detection method | | | Number of cells | Number of lost cells | p-value |
|---|---|---|---|---|---|
| TH staining | | | 98 (93-107) | 13 (10-16) | < 0.0001 |
| GAL4 | | | | | |
| NP6510 | | | 10 | ns | |
| TH | | | 12 | ns | |
| R48B04 | | | 45 | ns | |
| 104 | | | 50 | ns | |
| HL9 | | | 76 | ns | |
| R58E02 | | | 78 | 8 (5-10) | 0.0035 |
| split-GAL4 | AD | DBD | | | |
| MB312B | R58-E02 | R10-G03 | 13 | ns | |
| MB043B | R58-E02 | R32-D11 | 11 | ns | |
| MB299B | R58-E02 | R37-E10 | 13 (11-18) | 7 (6-13) | < 0.0001 |
| MB315C | R58-E02 | R48-H11 | 10 (7-12) | 2 (1-3) | 0.0006 |
| MB441B | R30-G08 | R48-B03 | 11 (8-12) | 2 (1-3) | 0.0006 |

mutants did not show the same phenotype[33]. Thus, the *Clk* has a role in preventing DA neuron loss and locomotor deficits in a circadian clock-independent manner[33]. Clock-independent roles of *tim* and *per* genes in mitochondrial uncoupling and lifespan have also been reported[60]. It is worth noting that MB299B neurons are not the first to degenerate in the basal conditions in *per⁰* at least up to day 7, as these neurons are present at the age when Ca²⁺ imaging was performed. These findings

underscore that clock genes play multiple roles in dopaminergic neuroprotection and are consistent with the notion that circadian disruptions are frequently comorbid with PD.

Short-term $H_2O_2$ treatment causes persistent increases in nighttime sleep but not locomotor defects. Blocking synaptic output of PAM-α1 neurons using *Shi^{ts}* increases daytime sleep and also decreases activity levels. Although sleep phenotypes observed in these two

**Fig. 5 | Identification of the vulnerable subpopulations of PAM neurons.**
**a, b** Quantification of the number of neurons labeled by GFP driven with different PAM neuron drivers, 7 days after a 4-h 10% $H_2O_2$ treatment at ZT20. The controls were treated with water only. **a** *UAS-EGFP* was driven with classical *GAL4* drivers as indicated in the *x*-axis. Among the drivers tested, only the *R58E02*-labeled neurons include a subpopulation vulnerable to oxidative insults. **p = 0.035 (two-tailed *t*-test). *n* = 7–25 hemispheres (see Source Data for individual sample numbers). **b** PAM subpopulations were labeled with split-*GAL4* drivers. Neurons labeled by MB299B, MB441B, and MB315C contain vulnerable PAM subpopulations. ***p < 0.001 (p = 0.0009 MB441B control vs. $H_2O_2$; p = 0.0004 MB315C control vs. $H_2O_2$) and ****p < 0.0001 (two-tailed *t*-test). *n* = 8–98 hemispheres (see Source Data for individual sample numbers). **c** Summary of the results from (**a**) and (**b**). The mean number of cells labeled by the given driver and of cells lost following the $H_2O_2$ treatment are shown. The numbers in the brackets indicate the range of values. AD activation domain, DBD DNA-binding domain. **d** Schematic representation of PAM subpopulations vulnerable to oxidative stress. PAM-α1, -γ5, and -γ3 neurons are labeled by MB299B, MB315C, and MB441B, respectively, with three PAM-α1 neurons co-expressing MB299B and MB315C (left). $H_2O_2$ treatment selectively degenerates approximately half of the PAM-α1 neurons and a few cells each from the PAM-γ5 and -γ3 subgroups (right). **e, f** 4-h 10% $H_2O_2$ treatment was performed at ZT8 and ZT20, and the number of MB299B-positive neurons was quantified 7 days post-treatment. **e** Representative confocal images of MB299B>EGFP neurons. Scale bar, 20 μm. **f** Both experimental groups have significantly fewer remaining neurons than the control group treated with water only at ZT20. $H_2O_2$ treatment at ZT20 caused a greater loss of neurons than at ZT8. *** p < 0.001 (p = 0.0001 control vs. ZT8 $H_2O_2$; p = 0.0003 ZT8 vs. ZT20 $H_2O_2$) and **** p < 0.0001 (two-tailed *t*-test). Control, *n* = 37; ZT20 $H_2O_2$, *n* = 98, ZT8 $H_2O_2$, *n* = 14 hemispheres. **g** In DD, 10% $H_2O_2$ treatment performed at CT4 and CT16 both led to a significant loss of MB299B neurons compared to the water-only control treatment at CT4. ****p < 0.0001 (two-tailed Mann–Whitney test). The number of remaining MB299B neurons was similar between CT4 and CT16 treatments. Control, *n* = 13; CT4 $H_2O_2$, *n* = 55, CT16 $H_2O_2$, *n* = 71 hemispheres. Source data are provided as a Source Data file.

experiments are not identical, the results collectively point to a role for PAM-α1 neurons in sleep regulation (without excluding other PAM subpopulations). Consistent with our finding, a previous study has shown that activation of PAM neurons reduces sleep[61]. Lack of loco-motor defects was expected, as another PAM neuron subset projecting to the MB β' lobe is known to control startle-induced climbing ability[14]. It is well established that PAM-α1 neurons play an essential role in the acquisition and consolidation of appetitive long-term memory[17,62] and the formation of courtship conditioning memory, which is a type of short-term memory[58]. Our findings that the circadian pacemaker cir-cuit controls these neurons raise the possibility that these learning/ memory performance types may show rhythmic variations across the day. Previous studies have indeed reported such phenomena[63,64], albeit with some disputes[65]. Moreover, because they are highly susceptible to oxidative stress-induced degeneration, their learning/memory func-tioning may be affected by oxidative stressors and are likely to be impaired when PAM-α1 undergoes degeneration. Thus, the circuit involving LNd−PAM-α1−MB α1 subdomain has overlapping roles in sleep and learning/memory and is susceptible to oxidative stress. Impairments in this pathway might elicit phenotypes that are remi-niscent of non-motor symptoms of PD.

Several lines of evidence support the multiple-hit hypothesis for PD, suggesting that progressive PD pathology is triggered by the interaction of multiple genetic/environmental risk factors[53]. Our find-ing that a single $H_2O_2$ treatment triggers DA neuron loss but that the rate of degeneration after that is not higher than the natural age-dependent loss in non-treated flies is consistent with the theory. We could not determine whether $H_2O_2$ treatment on *per⁰* accelerates dopaminergic neurodegeneration, as this combination causes pre-mature death. Of note, whereas immediate lethality has been reported when flies were treated with much lower concentrations of $H_2O_2$[31], we did not detect any such effect. Differences in a laboratory environment or fly strains might have caused this discrepancy. If not treated, PD presents a higher mortality risk, including sudden unexpected death[66]. Our findings are consistent with the multiple-hit hypothesis and sug-gest that genetic variations in circadian clock genes might represent a risk factor for dopaminergic neurodegeneration. We also demon-strated that an oxidative stressor administered at a specific time of day could have a critical impact on the survival of DA neurons and cause persistent changes in behavior; these findings might also be relevant in humans.

## Methods

### *Drosophila* strains and culture
Flies were reared in 12-h LD cycles in a humidified chamber at 25 °C on a standard corn meal medium. *Clk856-GAL4*[38] was a gift from Ralf Stanewsky. The following lines were previously described: *HL9-GAL4*[67], *R58E02-GAL*[16], *per⁰¹,w*[68], *w;tim⁰¹⁶⁹ Pdf-GAL4*[70], *Clk1982-GAL4*[38], *DvPdf-*

*GAL4*[41] *UAS-per16*[71], and *UAS-Shibire*[ts72]. *NP6510-GAL4* (113956) was obtained from the Kyoto Stock Center. The following lines were obtained from the Bloomington Stock Center: *TH-GAL4* (Stock number 8848), *R48B04-GAL4* (50347), *O104-GAL4* (62639), *MB312B* (68314), *MB043B* (68304), *MB299B* (68310), *MB441B* (68251), *MB315C* (68316), *UAS-GCaMP7s* (79032), *UAS-Kir2.1* (6596), *UAS-Catalase* (24621), *w¹¹¹⁸* (5905), *Canton-S* (64349), *UAS-myrGFP, QUAS-mtdTomato(3xHA)*, and *trans-Tango* (77124).

### $H_2O_2$ treatment
Male flies were collected after hatching and entrained to 12 h/12 h-LD cycles in an incubator at 25 °C. Seven-day-old flies were transferred to an empty vial for 5 h, and then a filter paper soaked with 100 μl $H_2O_2$ (Sigma Aldrich, 216763) of a given concentration or $H_2O$ was inserted in the vial for a defined duration. Flies were then placed in a vial con-taining standard corn meal agar food for 7 days before the analysis of DA neuron integrity. For circadian $H_2O_2$ treatment in LD, newly hat-ched flies were entrained to LD cycles, and on day 7, flies were treated with 5% or 10% $H_2O_2$ for 4 h at one of the 6 ZT time points (ZT0, 4, 8, 12, 16 and 20) following a 5-h food and water deprivation. After the $H_2O_2$ exposure, flies were placed in the vial with the standard food and maintained under LD cycles at 25 °C for 7 days. For circadian treatment in DD, newly hatched flies were entrained to LD cycles for 3 days and then released to DD. After three DD cycles, flies were treated with $H_2O_2$ at one of the CT time points (CT0, 4, 8, 12, 16 and 20), as described above. Treatment and subsequent incubation were performed in DD.

### Immunohistochemistry
Immunostaining was performed on whole fly brains. Flies were decapitated, and the whole heads were fixed in 4% paraformaldehyde + 0.3% Triton X-100 on ice for 1 h. The heads were washed with 0.3% Triton X-100 in phosphate-buffered saline (PBT) three times at room temperature. The head cuticles were partly opened, and the heads were incubated in a blocking solution (5% normal goat serum in PBT) for 1 h at room temperature. The incubation with primary antibodies was performed over two nights at 4 °C. Then, the heads were washed three times and incubated overnight with secondary antibodies at 4 °C. After three washes, the brains were dissected entirely by removing the remaining cuticle and trachea and mounted in Vectashield mounting medium (Vector laboratories, H-1000-10). The slides were stored at 4 °C and protected from light until imaging. The primary antibodies and the concentrations were as follows: rabbit polyclonal anti-TH (Millipore, ab152) 1:100; mouse monoclonal antibody nc82 (Develop-mental Studies Hybridoma Bank) 1:100; polyclonal rabbit anti-GFP (Invitrogen, A6455) 1:500; and monoclonal mouse anti-HA (Covance, MMS-101P) 1:200. The secondary antibodies and their concentrations were as follows: Alexa Fluor anti-rabbit 488 (ThermoFisher, A21052) 1:250 and Alexa Fluor anti-mouse 633 (ThermoFisher, A11008) 1:250.

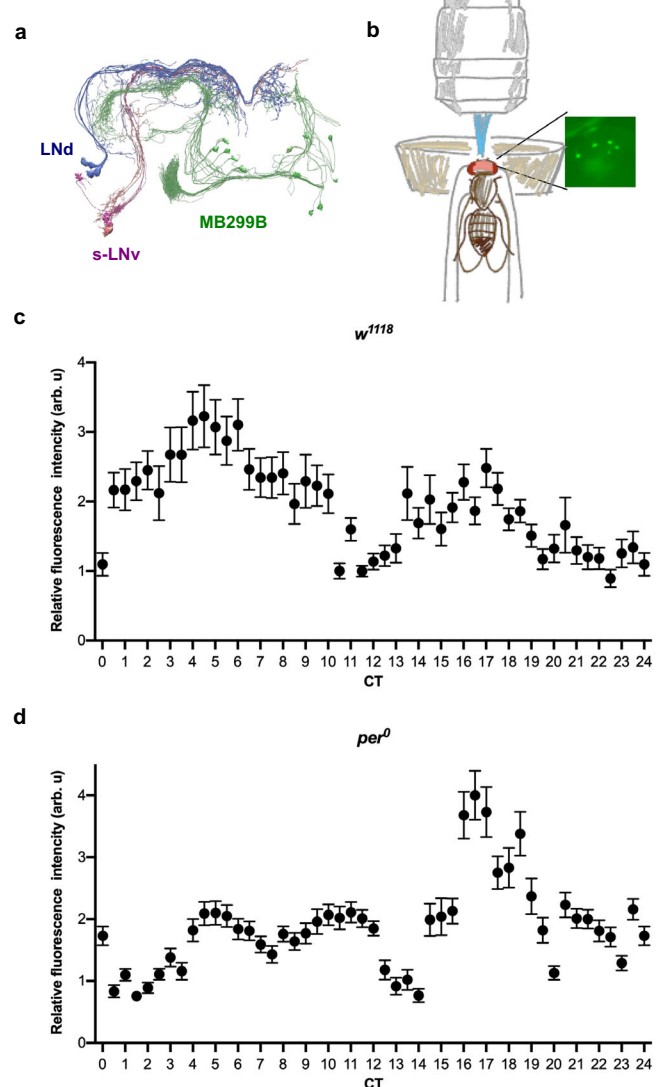

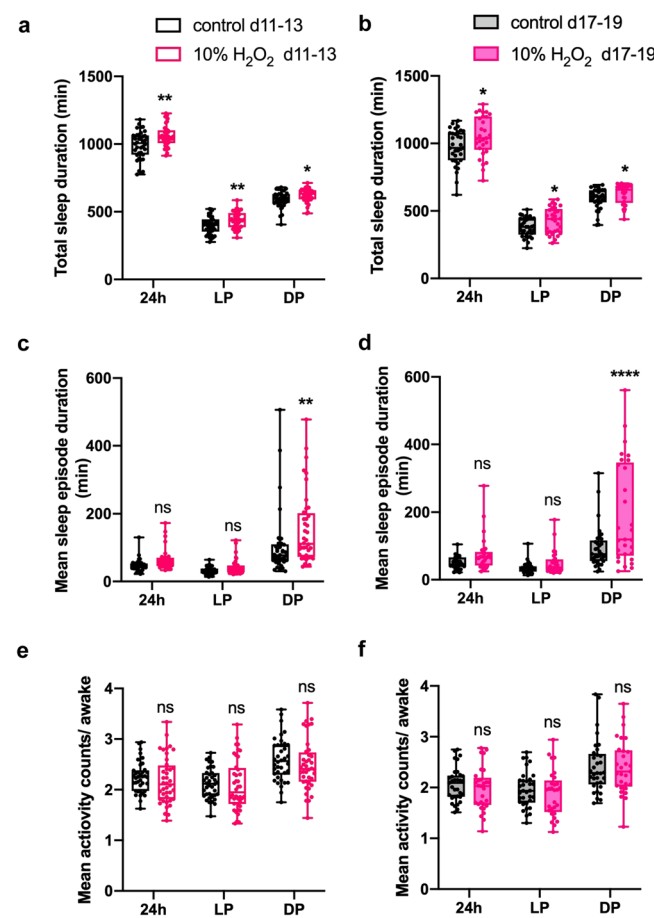

**Fig. 6 | MB299B neurons display Ca²⁺rhythms. a** Projections of the LNds contact dendritic arbors of MB299B neurons. Image created using the hemibrain connectome data[46] via the NeuPrint tool[47]. **b** Illustration of the method for live GCaMP imaging. **c, d** *MB299B > GCaMP7s* fluorescence levels in MB299B neurons throughout 24 h on the first day in DD following LD-entrainment in *w¹¹¹⁸* (**c**) and *per⁰* (**d**). Relative fluorescence intensity (mean ± SEM) was determined from three independent experiments. In (**c**), the analysis included an average of 79 (53–146 cells) cells per timepoint from 5–10 *w¹¹¹⁸* flies (see Source Data for individual sample numbers per timepoint). In (**d**), an average of 76 (44–131) cells per timepoint from 5–10 *per⁰* flies were analyzed (see Source Data for individual sample numbers per timepoint). Source data are provided as a Source Data file.

**Fig. 7 | A single short-term H₂O₂ treatment increases nighttime sleep.** Seven-day-old *w¹¹¹⁸* flies were treated with 10% H₂O₂ or water for 4 h starting at ZT20 in LD and then placed in the activity monitor. The sleep and activity of flies from age day 11 to 13 (d11-13) (control, *n* = 38; H₂O₂, *n* = 40 flies) and day 17 to 19 (d17-19) (control, *n* = 33; H₂O₂, *n* = 29 flies) were analyzed. **a, b** Total sleep duration over 24 h (24h), during daytime (light period, LP) and the night (dark period, DP) in water-only control and H₂O₂-treated flies from day 11 to 13 (**a**) and day 17 to 19 (**b**). H₂O₂-treated flies show an increase in total sleep in both age groups. **p* < 0.05 and ***p* < 0.01 (two-way ANOVA with Šídák's multiple comparisons test). **c, d** Mean sleep episode duration from day 11 to 13 (**c**) and day 17 to 19 (**d**). H₂O₂-treated flies show an increase in nighttime sleep episode duration in both age groups. ***p* < 0.01 and *****p* < 0.0001 (two-way ANOVA with Šídák's multiple comparisons test). **e, f** Mean activity counts during the wake period from day 11 to 13 (**e**) and day 17 to 19 (**f**). No significant difference was observed between the control and treated groups (two-way ANOVA with Šídák's multiple comparisons test). Source data are provided as a Source Data file.

clear the debris. The absorbance of the supernatant was measured at 625 nm on a spectrophotometer. The measured value was divided by the number of flies homogenized in one tube.

### ROS detection
MitoSOX Red (Invitrogen, M36008), a cell-permeable mitochondrial superoxide indicator, was used to measure endogenous ROS levels. Male flies were decapitated, and brains were dissected in Hanks' Balanced Salt Solution (HBSS), followed by incubation in 5 µM Mito-SOX for 15 min at 37 °C and three washes with warm HBSS. The brains were mounted in HBSS and imaged immediately with a Leica TCS SP5 confocal microscope or a Nikon Ax confocal microscope.

### *trans*-Tango
The *trans*-Tango assay was performed as described previously[43]. Briefly, male flies were aged for 10 days at 18 °C to maximize *trans*-

### Feeding assay
The feeding assay was performed as previously described with minor modifications[35]. Male flies were starved for 5 h before the assay (beginning of feeding). A filter paper soaked with a 0.125 mg/ml solution of Brilliant Blue FCF was then inserted in the vial for a control group. In the viral containing the flies of the experimental group, a filter paper soaked with a 0.125 mg/ml solution of Brilliant Blue FCF with 10% H₂O₂ was inserted. The filter paper was kept in the vials for 4 h, after which the flies were flash-frozen in liquid nitrogen. Then, the flies were vortexed to detach their heads from bodies. A standard sieve (No. 25) was used to remove heads, and the bodies were transferred to 1.5-ml Eppendorf tubes. The bodies were homogenized with a motorized pestle in 1 ml of PBT and centrifuged at 13,000 rpm for 15 min to

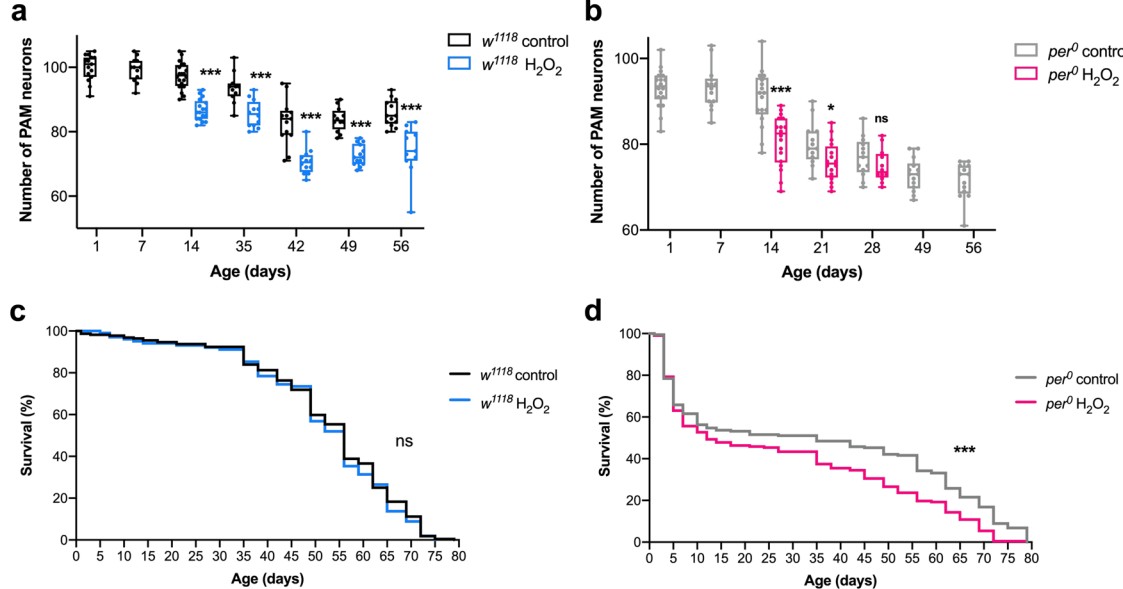

**Fig. 8 | A single short-term H₂O₂ treatment causes premature death in *perᵒ* flies.**
PAM neuron counts and survival of *w¹¹¹⁸* and *perᵒ* flies following a 4-h exposure to
10% H₂O₂ at ZT20 when flies were 7 days old. **a**, **b** PAM neuron counts across aging
in *w¹¹¹⁸* (**a**) and *perᵒ* (**b**). PAM neuron loss was observed as early as day 14 in both
genotypes compared to control groups treated with water only. Neurodegenera-
tion progresses with age but is not accelerated in H₂O₂-treated flies. *$p = 0.02$ and
***$p < 0.001$ (two-tailed *t*-test). $n = 12-20$ hemispheres (see Source Data for indivi-
dual sample numbers). Since H₂O₂-*perᵒ* flies exhibited a high mortality rate after
28 days, PAM neurons of this group were not examined thereafter. **c**, **d** % of

surviving flies in *w¹¹¹⁸* (**c**) and *perᵒ* (**d**). H₂O₂ treatment did not affect the survival of
*w¹¹¹⁸* flies, whereas H₂O₂-treated *perᵒ* flies had significantly reduced lifespans com-
pared to the water-treated control group. ***$p = 0.0002$ (Log-rank test). In (**c**), $n = 3$
independent experiments for both control and H₂O₂ treatment, with a total of
$n = 324$ *w¹¹¹⁸* flies for control and $n = 262$ for H₂O₂ treatment. In (**d**), $n = 3$ indepen-
dent experiments for both the control and the H₂O₂-treated groups, with a total of
$n = 190$ *perᵒ* flies for the control and $n = 203$ for the H₂O₂-treated group. Source data
are provided as a Source Data file.

Tango expression with the optimal signal-to-noise ratio. The immu-
nohistochemistry protocol described above was used to visualize the
neurons and their projections.

### In vivo calcium imaging

The calcium sensor GCaMP7s[48] was expressed with MB299B split-
GAL4, and the 3-day-old flies were entrained to LD cycles for 4 days. On
the first day in DD following the LD entrainment, flies were placed
under the fluorescence microscope in the perfusion chamber after the
brief operation to remove a small part of the cuticle of the head cap-
sule. Each experiment consisted of a session of 6-h to 8-h live imaging
at 30-min intervals, starting at four different circadian times (CT4,
CT10, CT16, and CT22). During each session, 4 flies were recorded, and
on average, 13 (ranging from 5 to 19) GFP-positive neurons were
imaged per fly. Images were acquired using the Leica DM550 fluor-
escent microscope. The exposed brains were constantly perfused with
oxygenated saline to prevent desiccation or deterioration of the tissue
during imaging. Throughout the experiment, the flies retained the
ability to move their abdomen and legs. The death of a fly was deter-
mined by the absence of movement. Additionally, a sudden drop in
fluorescence intensity and a rapid increase in background auto-
fluorescence were considered indicators of tissue deterioration. Data
from the flies that died or displayed signs of deterioration during the
experiment were excluded from the analysis. The fluorescence inten-
sity of individual cells was measured using the freehand selection tool
of Fiji/ImageJ. The corrected total cell fluorescence (CTCF) value was
obtained by subtracting the measured fluorescence value outside of
MB299B neurons from the measured fluorescence value inside
MB299B neurons. The following formula was applied to calculate
CTCF: CTCF = integrated density − (area of the region of interest x
mean fluorescence of background). The mean of the CTCF of indivi-
dual cells per timepoint was calculated, and the normalized CTCF time

series was obtained by dividing the mean CTCF of each timepoint by
the lowest mean CTCF over 24 h.

### Sleep assay and data analysis

The fly sleep assay was performed as described previously with minor
modifications[27]. Seven-day-old male flies were deprived of food and
water for 5 h and treated with 10% H₂O₂ for 4 h starting at ZT20. The
control group was treated with water only. After the treatment, flies
were placed in individual glass tubes containing 5% agarose with 2%
sucrose in the *Drosophila* Activity Monitoring (DAM) System 3 (Triki-
netics) and entrained in 12 h:12 h-LD cycles at 25 °C for 3 days. Loco-
motor activity data were recorded during the subsequent 12 days in
LD. We extracted sleep quantity from the activity data collected in
1-min bins (defined as bouts of inactivity lasting for 5 min or longer)[73].
Sleep analysis was performed in MATLAB (MathWorks, version
R2017a) using SCAMP v2 (Sleep and Circadian Analysis MATLAB Pro-
gram) according to the instruction manual (Vecsey laboratory)[74].
*MB299B>Shiᵗˢ* and *UAS-Shiᵗˢ* flies were raised at 19 °C in 12 h:12 h-LD
cycles until 7 days old and placed in the activity monitor at 19 °C,
12 h:12 h-LD cycles. When they were 11 days old, the temperature was
shifted to 30 °C until the end of the experiment. Locomotor activity
and sleep data between day 11 and 13 were analyzed as
described above.

### Startle-induced negative geotaxis assay

The startle-induced negative geotaxis assay was performed as pre-
viously described[75] with minor modifications[21]. Twenty male flies were
anesthetized with CO₂ and placed in a 100-ml graduated glass cylinder.
The cylinder was divided into five equal zones and graded from 1 to 5
from the bottom to the top. The bottom of the cylinder was marked as
zone 0. After 1 h of recovery from the CO₂ exposure, the climbing assay
was performed at ZT2. Flies were gently tapped to the bottom of the

column to startle them, from which they recovered quickly and started climbing the cylinder wall. The whole experiment was video-recorded for subsequent analyses. The number of flies that climbed up to each zone within the 20 s after startling was manually calculated, from which climbing index (CI) was calculated using the following formula: CI = $(0 \times n_0 + 1 \times n_1 + 2 \times n_2 + 3 \times n_3 + 4 \times n_4 + 5 \times n_5)$ / $n_{total}$, where $n_{total}$ is the number of total flies used in the experiment and $n_x$ is the number of flies that reached the zone X. Three trials for each experimental group were performed at 30-s intervals, from which the median values were chosen as final data of an experiment. Three to four independent experiments were performed per group per age.

### Lifespan assay
Lifespans of flies were assayed as previously described with minor modifications[76]. Briefly, 1-day-old male flies were placed in the vials containing cornmeal agar food, 10 flies per vial, and maintained under 12 h:12 h-LD cycles at 25 °C. At 7 days old, treatment groups were exposed to 10% $H_2O_2$ for 4 h starting at ZT20, as described above. Control groups were treated with water only. Following the treatment, flies were returned to the vials with food. Flies were transferred to new vials twice a week, and the number of dead flies was recorded daily throughout the experiment.

### Confocal microscopy and image analysis
Fly brains were imaged using a Leica TCS SP5 confocal microscope and a Nikon Ax confocal microscope. All images were analyzed using Fiji/ImageJ software (NIH). To count DA neurons, neurons positive for TH immunostaining or neurons labeled with fluorescent protein were counted manually using the cell-counter plugin of ImageJ through individual Z-stacks of the confocal images. To quantify the fluorescence intensity of the MitoSOX Red in the region of PAM neurons expressing GFP, the measurement feature of ImageJ was used after adjusting a proper threshold. To calculate the resulting fluorescence intensity of the MitoSOX Red in the region of PAM neurons, the CTCF value was obtained by subtracting the measured fluorescence value outside of the PAM region from the measured fluorescence value inside PAM neurons.

### Statistical analysis.
GraphPad Prism (v.8.1) and R were used for statistical analysis and data plotting. All data were assessed using the D'Agostino-Pearson normality test. No statistical method was used to predetermine the sample size. Sample sizes were chosen without calculation but based on the literature describing similar experiments. Investigators were not blinded to genotypes and conditions in most experiments because only one experimenter has conducted the entire course of these experiments. However, all data were analyzed using unbiased statistical methods. Experiments performed by multiple investigators (Figs. 2a–d, 3d, e, 4c, 8c, d, and Supplementary Figs. S2a, b, and S7a–c) were analyzed blindly to the genotype and conditions. Normally distributed data were compared using parametric tests, and non-normally distributed data were analyzed using nonparametric tests. To compare the two groups, the two-tailed Student's $t$-test was used for normally distributed data, and for non-normally distributed data, the nonparametric Mann–Whitney test was used. To compare three or more groups of normally distributed data, one-way analysis of variance (ANOVA) with post hoc Tukey's multiple comparison test and two-way ANOVA with Šídák's multiple comparisons test were used. The Kruskal–Wallis one-way ANOVA with Dunn's multiple comparison test was used to compare non-normally distributed data. The statistical significance threshold for all experiments was set at $p < 0.05$. In all figures, stars represent statistical significance: * $p < 0.05$, ** $p < 0.01$, *** $p < 0.001$, and **** $p < 0.0001$. ns indicates not significant. Micrographs are representatives of two or more independent experiments. The box boundaries of all box plots are the 25th and 75th percentiles,

the horizontal line across the box is the median, and the whiskers indicate the minimum and maximum values. All data points are displayed.

### Reporting summary
Further information on research design is available in the Nature Portfolio Reporting Summary linked to this article.

## Data availability
All data are available in the main text or the supplementary materials. Source data are provided with this paper.

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

## Acknowledgements

We thank Dong-gen Luo for helpful technical advice and the Bloomington *Drosophila* Stock Center and Kyoto Stock Center for fly strains. We also thank Ralf Stanewsky for a fly strain and Serge Birman for comments on the manuscript. The following funding sources supported this research: Swiss National Science Foundation grants to E.N. (31003A_169548 and 310030_189169), Georges and Antoine Claraz Foundation to E.N., Ernst and Lucie Schmidheiny to EN Société Académique de Genève to E.N.

## Author contributions

Conceptualization: M.M.D. and E.N. Investigation: M.M.D., L.C.D., and E.P. Visualization: M.M.D., L.C.D., E.P., and E.N. Writing: M.M.D., L.C.D., and E.N.

## Competing interests

The authors declare no competing interests.
