## [Peer Review File · Nature Communications]

Circadian clock disruption promotes the degeneration of dopaminergic neurons in male *Drosophila*REVIEWER COMMENTS

Reviewer #1 (Remarks to the Author):

This is an interesting and clearly presented study aimed at determining how the circadian clock impacts dopaminergic (DA) neuron degeneration. This is an important topic, since there is a strong link between Parkinson's disease and circadian rhythms and sleep disruptions. However, the mechanisms underlying this connection are poorly understood, and how the circadian clock impacts DA neurons remains to be established.

The authors cleverly used H₂O₂ exposure to explore how the circadian clock regulates DA neurodegeneration. They identify diurnal and circadian (per-dependent) rhythms of sensitivity to H₂O₂-induced degeneration in a group of DA neurons: the PAM neurons. They focus on a small subpopulation within the PAM neurons that is particularly sensitive to H₂O₂ exposure, and show that these neurons are post-synaptic to specific circadian neurons. A functional clock is required in circadian neurons to protect PAM neurons from degeneration but how they do so was not established, though neuronal activity was excluded as a mechanism. Finally, the authors provide evidence that PAM neuron loss is associated with increased sleep, and also note that loss of circadian rhythms and exposure to H₂O₂ cause premature death. These results, combined with previous work, indicate that *Drosophila* could be a very potent model to understand the connection between circadian clocks and Parkinsonian neurodegeneration, and reveal the importance of circadian neurons in modulating DAergic neuron degeneration. There are however significant issues that should be addressed:

Major comments:

- 1) The author found that sleep is increased when flies are exposed to H₂O₂ but they did not establish that the loss of PAM- α 1 neurons is the cause. Have PAM- α 1 been previously shown to control sleep? Does inhibition of this neurons alter sleep patterns? A supporting experiment might be to compare sleep loss in three cohorts of flies: a) flies in which PAM- α 1 neurons are inhibited or ablated, b) flies treated with H₂O₂, and c) flies with inactivated or missing PAM- α 1 treated with H₂O₂. No additive effect on sleep amount would be expected with H₂O₂ if indeed loss of PAM- α 1 neurons explain the sleep defects observed with H₂O₂.
- 2) The model proposed by the authors is that circadian rhythms control rhythmic sensitivity of DA neurons to H₂O₂. The manuscript would be strengthened by showing that rhythmic H₂O₂ sensitivity is rescued when pacemaker function is restored in circadian neurons of per null flies. The authors showed rescue only at a single time point.
- 3) It is shown on figure 2D that CS and w1118 strains exhibit differences in vulnerability to H₂O₂-induced neurodegeneration. Thus, ideally, experiments would be performed with flies backcrossed to a standard background such as w1118. Did the authors do this, at least for per0? Since the authors were able to observe rescue of sensitivity, the risk of a background effect explaining the per0 phenotype is reduced. However, they did not report on figure 4A/B the results with per0;uas-per flies. Since all genotypes presented showed rescue or at least a trend toward improvement, this control should be included.
- 4) The MitoSox experiment in figure 3C does not include a measurement at CT4, the time point at which vulnerability to H₂O₂ was the strongest. The authors might thus have missed an increase in ROS levels at this time point that could underlie the DD rhythm of PAM H₂O₂ sensitivity. This is especially significant as the authors use this result to conclude that "another mechanism [other than ROS rhythms] under the control of circadian clocks gates the vulnerability of PAM neurons in DD." (line 195-6). This is the only explanation provided for the rather surprisingly different phases of H₂O₂ sensitivity observed between LD and DD in control flies. Also surprising is the very different phase observed under LD in per0. If light was driving oscillations in LD in control flies, why does per0 show such a different phase compared to wild-type flies?
- 5) The mapping of sensitive DA neurons to a very specific subgroup is really interesting. However, it was done in LD. Is it possible that different neurons might be sensitive under LD and DD, and that this would explain phase differences? It would be worth repeating the experiments in DD for at least MB299B neurons.
- 6) Sleep disruptions are emphasized as a symptom of neurodegenerative disorders and Parkinson's disease. It would be worth determining whether sleep consolidation is affected after

H2O2 treatment.

Minor:

- 1) Line 74, I would use plural for the MBs (bodies instead of body).
- 2) Line 89: Writing that PAM-a1 are "selectively" susceptible might be too strong since two other groups of PAM neuron show some sensitivity.
- 3) Rephrase line 146.
- 4) Line 172-175: Why do flies drink so much more in DD?
- 5) Paragraph starting at line 197: I presume the clkGAL4 driver used here is also expressed in glia and other tissues. I understand that later the authors focus on circadian neurons, but I would suggest mentioning in this paragraph that the driver is not just expressed in clock neurons, and modifying the conclusion on line 204,
- 6) Line 245: when introducing split-GAL4 drivers, it would be worth indicating which regions of the MB each innervate.
- 7) Line 286: Ca²⁺ peaks occur at CT5 and CT17. They would thus be better described as "just before midday and midnight", rather than "morning and early night"
- 8) Line 287: in the sentence "Because Ca²⁺ levels in neuronal cell bodies can be correlated with firing frequency, we wanted to know whether decreased excitability would be neuroprotective" is unclear. This does not seem to flow logically: how does the association between firing frequency and Ca²⁺ levels lead to questioning if excitability was associated with increased vulnerability?
- 9) Figure 2A/B: Is the "number of PAM neurons," represented in 2B, cells that co-stain with Red Stinger and GFP, or just Red Stinger-positive cells? Also, the representative images chosen for the per0 brains in 2A do not seem to correlate or represent well the data presented in 2B. In particular, the number of RedStinger-positive cells at Day 1 seems to be much lower than at day 7 or day 14. I must say the image quality is not very good in the file we received, so perhaps some of the staining is obscured by the poor resolution.
- 10) Figure 4D: although the PDF- 5th sLNv is included in the text as one of the neurons that could project to the PAM neurons, the diagram does not depict it sending projections to the PAM neurons.
- 11) Figure 5B: MB441B should be moved to the rightmost panel (switched with MB315C) to match the order in which these drivers are introduced in the text.
- 12) Figure 5. It would be preferable to keep the order of GAL4 and split-GAL4 drivers consistent throughout the figure.
- 13) Figure 5E: the representative images do not correlate with the quantification in 5F. For the images depicting H2O2 exposure at ZT8 and ZT20, it appears that both have about 8-10 neurons left, in contrast with 5F.

Reviewer #2 (Remarks to the Author):

The study by Dorcikova et al. investigates whether Parkinson's disease (PD) is associated only with disturbances in sleep and circadian rhythms or whether these disturbances are causally responsible for PD, as they often precede the actual PD symptoms. To answer this question the authors used the well-established model *Drosophila melanogaster*, in which the PD typical loss of dopaminergic (DA) neurons can be induced by oxidative stress (e.g. feeding the flies H₂O₂). The group of Emi Nagoshi has previously shown that H₂O₂ feeding causes the loss of specific DA neurons in the PAM cluster and that this leads to defective climbing behavior suggesting a partial analogy between the PAM cluster and the mammalian substantia nigra. The PAM neurons comprise ~130 quite heterogenous neurons that, in addition to locomotion, play critical roles in olfactory associative learning, foraging and sleep.

To test for circadian rhythms in vulnerability to oxidative stress, the authors developed a protocol for short-term H₂O₂ treatment (4h of 5% or 10% H₂O₂) that can be administered at different times of day. They show that this treatment leads reliably to the loss of specific PAM neurons and that the loss depended on the time of day. More importantly, they demonstrate that the rhythm in vulnerability to oxidative stress is controlled by the circadian clock since it persists in the absence of external Zeitgebers and is absent in arrhythmic per0 mutants. Moreover, they show that

arrhythmic mutants (*per0* and *tim0*) possess fewer DA neurons in the PAM cluster even without H₂O₂ treatment and that they lose more DA neurons in response to oxidative stress. Thus, the circadian clock controls the magnitude and temporal sensitivity of the PAM neurons to oxidative stress. Subsequently, the authors performed several critical control experiments that confirmed that it is indeed the circadian clock in specific clock neurons (the PDF-positive LN_vs) that regulates the survival of PAM neurons after oxidative stress. Furthermore, they identified the specific PAM neurons (PAM- α 1) that were predominantly killed by oxidative stress and showed that the LN_d clock neurons are presynaptic to them. They demonstrate that the PAM- α 1 neurons show bimodal rhythms in intracellular Ca²⁺, which are absent in *per0* mutants and show that the loss of these neurons affects sleep but not climbing behavior. Finally, the authors investigated the interplay between genetic factors, environmental risk factors and age on the loss of DA neurons in the PAM cluster and found that all factors contribute. This confirms the multi-hit hypothesis for developing PD.

Overall, the study is carefully conducted and clearly shows that disruption of the circadian clock promotes dopaminergic neurodegeneration in flies. The results obtained suggest that mutations in circadian clock genes are an additional risk factor for the development of PD, along with environmental factors and age. These findings could also be relevant for humans and significantly expand our knowledge of the causes of PD.

Nevertheless, the study has several weaknesses that need to be improved before publication. In some places, relevant literature is not cited or discussed and alternative hypotheses are not mentioned. Furthermore, the methods used are not always adequately described.

Specific comments:

Lines 61-72: citing the review of Hall from 2003 and an own review from 2019 in respect of the clock neurons and the clock network is certainly fine, but what about the recent work of Shafer et al. (2022, *eLife* 11) and Reinhard et al. (2022, *JCN* 530 and *Front Physiol* 19) that describe the clock network and their synaptic connections in greater detail? This work needs to be cited.

Lines 72-74: again, only the authors' own work is cited. What about the work of Liang et al. (2019, *Neuron* 102) showing that the circadian clock controls ring neurons in the ellipsoid body via PPM neurons? Since PPM neurons are also dopaminergic neurons, this work is very relevant to the present study (see also later).

Lines 139-143, Figure 2 and Material and Methods: *w1118* and *CantonS* (CS) flies were used as controls for *per0* flies, but none of these stocks is listed in M & M. I suppose that *per0* flies are red eyed, thus CS flies are probably the better controls. Do you have an explanation for why the *w1118* flies are as sensitive to 10% H₂O₂ as the *per0* mutants (Fig. 2D)?

Figure 2: Are the images in 2A superpositions of several confocal stacks or do they show only one confocal image? How thick are the individual confocal images? The image of the 1-day-old *per0* fly appears to contain fewer stinger-positive PAM neurons than the images of the older flies, but based on the quantification in Fig. 2C, this is not representative. The many small symbols (data points) in the box plots make it hard to distinguish the white (*w1118*) and gray (*Canton-S*) plots. Please try to optimize the pictures. In my opinion, it is enough to indicate the number of analyzed brain hemispheres below the box plots.

Lines 199-205: most people use the *Clk856-gal4* line for driving expression in all clock neurons. Please explain briefly why you used *Clk1982-gal4* for rescuing *per* in all clock neurons. Are there any differences between the two lines?

Line 208: the *Pdf-gal4* line was not generated by Renn et al. (1999) but by Park et al. (2000, *PNAS* 97). Renn et al. did also not show that the fifth s-LN_v does not express PDF. The fifth s-LN_v was first described as PDF negative by Kaneko et al. (1997, *J Neurosci* 17), whereas Rieger et al. (2006, *J Neurosci* 26) first functionally characterized the fifth s-LN_v as an evening neuron. Please correct your citation.

Lines 266-269: What do you mean by visual inspection? Did the hemibrain connectome data reveal any synapses between the LNDs and MB299B neurons, and if yes, how many? You can even study Shafer et al. (2022), since these authors might have already reviewed this.

Figure 6 and Material and Methods: I did not completely understand how you generated Figure 6C and D. As you stated in Material & Methods Ca²⁺ imaging could be performed in a single fly for up to 6-8 hours and imaging was started every 6 hours to cover the entire day. How many flies could you record at once, and how many cells you imaged on average in each single fly? How similar were the fluorescence intensity values in different PAM neurons within the same fly? How did you detect the death of a fly, and how did you decide when to stop measurements on individual flies and discard the data? If I understood it right, at certain time points, the diagram for per0 mutants includes the measurements of only 2 flies. At how many time points was n=2? Could the low number of tested flies explain the irregular shape of the curve? In particular, between CT15 and CT18 there is a sudden jump to higher values of fluorescence intensity. Did this jump coincide with a lower number of flies tested? Would there be a possibility to color code the points so that one can immediately tell from how many flies was recorded?

Lines 364-366: what do you mean with "a synaptic structure that indicates fast neurotransmitter signaling was not detected between the LNDs and PAM- α 1 neurons in the connectome data"? In the results section, you considered it very likely that these two neurons have synaptic contacts with each other and you showed this even via trans-Tango. Now you say that there are no synapses between the two types of neurons. Please clarify this discrepancy. Nevertheless, the discussed paracrine, modulatory input from the clock neurons is very likely, since all of them are neuropeptidergic. I wonder why you exclude a neuropeptidergic input to the PAM- α 1 neurons from the s-LNVs via PDF and/or sNPF. The fibers of these two types of neurons are certainly close enough to enable paracrine signaling. At least for PDF this could be easily tested by cAMP-imaging in the PAM- α 1 neurons during the application of PDF. Furthermore, the presence of the PDF-receptor on the PAM- α 1 neurons could be checked using a PDFR-gal4 line. Since you could prevent the loss of PAM neurons by expressing PER only in the PDF neurons (Figure 4), a signalling from the PDF neurons to the PAM- α 1 neurons appears very likely. This is actually the simplest explanation for your findings.

Lines 372-388: The bimodal Ca²⁺ rhythms in the MB299-labelled neurons are very interesting. Similar bimodal Ca²⁺ rhythms have been found in ring neurons of the ellipsoid body neurons by Liang et al (2019, see above). These ring neurons are rhythmically modulated by DA neurons of the PPM3 cluster, which get neuromodulatory input from the PDF neurons. Thus, this situation strongly resembles the findings in your study and I do not understand why this is not discussed and the Liang et al (2019) paper is not even cited.

Lines 408-410: Lyons and Roman (2008 Learn Mem 16) have already shown that the circadian clock modulates short-term memory. Later, Chouhan et al (2015 Curr Biol 25) even showed that flies remember the time of day, an ability that requires the input from the circadian clock to memory centers in the mushroom bodies. Citing these papers may help to discuss the non-motor symptoms of PD.

Figure 8: The sudden death of all per0 mutants treated with H₂O₂ on day 31 is hard to understand. You state that this experiment was repeated twice. It is hard to believe that all flies died exactly at the same day in both experiments. How many flies are included in the survival assay? Have they been kept in just one single large food vial or several food vials? How often food was changed? I did not find any description of this experiment in Material & Methods. This description must be added, and I strongly recommend to repeat this experiment once more.

Minor points:

Line 117: Remove the fullstop after "post-treatment".

Lines 145-147: please change into: "...showed a significant loss of PAM neurons as compared to the control group..."

Lines 172-173: Just for interest: do you have an explanation for why flies drink three to five times more H₂O₂ in DD than in LD?

Reviewer #3 (Remarks to the Author):

Dorcikova et al lay out an interesting and exciting study examining links between circadian rhythm and loss of dopaminergic neurons as a potential mechanism occurring in Parkinson's. The manuscript is very well written and clear with not observable typos or missing details that I can see.

The study uses flies to initially recapitulate a loss of DA neurons in response to H₂O₂, noting a circadian influence in the effect depending on time of delivery of the insult. They then examine DA neuronal loss in flies mutant for circadian clock components, noting a loss of neurons in the *per0* mutant and a timeless mutant. Given that the chemical insult is delivered by feeding, they then examine any role for starvation or dehydration and find no role.

The key population of neurons are the PAM group and they examine synaptic linkage between circadian neurons and PAM neurons and find connections, data that is bolstered by the Janelia Farm connectome data. A subpopulation of the PAM neurons is then genetically defined and these are examined for their resting calcium levels in a 24h period, with a dramatic change in the calcium rhythms altered in the *Per0* mutant. This data is interesting, but more descriptive than mechanistic as it stands. The subpopulation of PAM neurons are associated with the mushroom body, and may regulate sleep. The paper then studies the effects of the H₂O₂ administration on sleep and finds an increase in nighttime sleep. I'm not so comfortable with the specificity of this experiment, as the H₂O₂ delivery is systemic, but the conclusions based on the loss of the subset of DA neurons – so the linkage is limited here.

I like this study a great deal and feel that it opens up a productive area of study. My concerns are with the linkage between the feeding and the effects, is ROS in either the DA neurons or the Pdf neurons eliciting these effects. I'm assuming that is the claim, but it would be good to link ROS, either the pdf or DA neurons more directly than the presence of increased MitoSOX detected ROS in the PAM neurons. I would suggest a rescue of the H₂O₂ excess in either the pdf neurons or the PAM neurons using UAS-catalase (or some other anti-oxidant transgene – UAS-Trx, UAS-GST?) to reduce ROS specifically in these neurons to define whether ROS toxicity is causing this effect in pdf neurons, or DA neurons.

Overall though, this is a meticulous and novel study and potentially lays out mechanisms for the two-hit theory of insult causing PD

Reply to Reviewers' comments

Comments from Reviewers are shown in blue font and our replies are in black.

Reviewer #1 (Remarks to the Author):

This is an interesting and clearly presented study aimed at determining how the circadian clock impacts dopaminergic (DA) neuron degeneration. This is an important topic, since there is a strong link between Parkinson's disease and circadian rhythms and sleep disruptions. However, the mechanisms underlying this connection are poorly understood, and how the circadian clock impacts DA neurons remains to be established.

The authors cleverly used H₂O₂ exposure to explore how the circadian clock regulates DA neurodegeneration. They identify diurnal and circadian (per-dependent) rhythms of sensitivity to H₂O₂-induced degeneration in a group of DA neurons: the PAM neurons. They focus on a small subpopulation within the PAM neurons that is particularly sensitive to H₂O₂ exposure, and show that these neurons are post-synaptic to specific circadian neurons. A functional clock is required in circadian neurons to protect PAM neurons from degeneration but how they do so was not established, though neuronal activity was excluded as a mechanism. Finally, the authors provide evidence that PAM neuron loss is associated with increased sleep, and also note that loss of circadian rhythms and exposure to H₂O₂ cause premature death. These results, combined with previous work, indicate that *Drosophila* could be a very potent model to understand the connection between circadian clocks and Parkinsonian neurodegeneration, and reveal the importance of circadian neurons in modulating DAergic neuron degeneration. There are however significant issues that should be addressed:

Thank you for the constructive comments and suggestions. We provide a marked-up copy of the revised manuscript together with a clean copy for your perusal.

Major comments:

1) The author found that sleep is increased when flies are exposed to H₂O₂ but they did not establish that the loss of PAM- α 1 neurons is the cause. Have PAM- α 1 been previously shown to control sleep? Does inhibition of this neurons alter sleep patterns? A supporting experiment might be to compare sleep loss in three cohorts of flies: a) flies in which PAM- α 1 neurons are inhibited or ablated, b) flies treated with H₂O₂, and c) flies with inactivated or missing PAM- α 1 treated with H₂O₂. No additive effect on sleep amount would be expected with H₂O₂ if indeed loss of PAM- α 1 neurons explain the sleep defects observed with H₂O₂.

A previous study has shown that activating PAM neurons reduces sleep (Nall et al. Sci Rep 2006 : now cited in the revised manuscript, lines 458-459). However, the specific role of PAM- α 1 neurons in sleep has not been demonstrated. We agree that, whereas our data show that the H₂O₂ treatment causes an increase in nighttime sleep, whether it is the consequence of PAM- α 1 loss was not directly tested. To address this issue, as suggested, we inhibited the synaptic output of PAM- α 1 neurons using the temperature-sensitive mutant of dynamin, *Shibire^{ts}*, and found that blocking PAM- α 1 output alters sleep and activity: increase in total and consolidation of sleep during the day, decrease in nighttime sleep, and reduction in activity levels throughout 24 h (new Fig. S7a-c). The phenotypes after the H₂O₂ treatment and blocking the output of PAM- α 1 are not identical, which could be due to the temperature difference (25°C for H₂O₂ treatment and 30°C for *Shi^{ts}* experiments) or incomplete blockage of neuropeptide release by *Shi^{ts}*. (It is reported that *Shi^{ts}* fails to inhibit output of some peptidergic neurons, such as PDF-positive LNvs. Ref. Mabuchi et al. 2016; Kilman et al. 2009). Thus, our results collectively point to the involvement of PAM- α 1 neurons in sleep regulation. This set of results is described in lines 353-366 and discussed in lines 455-458.

2) The model proposed by the authors is that circadian rhythms control rhythmic sensitivity of DA neurons to H₂O₂. The manuscript would be strengthened by showing that rhythmic H₂O₂ sensitivity

is rescued when pacemaker function is restored in circadian neurons of *per* null flies. The authors showed rescue only at a single time point.

As suggested, we have repeated the genetic rescue of *per⁰* in clock neurons and analyzed the vulnerability of PAM neurons at two timepoints in DD, CT4 and 16. We included a second pan-clock neuron driver, *Clk856-GAL4* in this assay. All the rescue genotypes prevented loss of PAM neurons at both time points. The new results are displayed in Fig. 4c and described in lines 227-239. Remarkably, *per* rescue with *Clk856-GAL4* or *Pdf-GAL4* resulted in the complete prevention of PAM neuron loss at both timepoints. This finding suggests that the overexpression of PER enhanced the resistance of PAM neurons.

3) It is shown on figure 2D that CS and *w¹¹¹⁸* strains exhibit differences in vulnerability to H₂O₂-induced neurodegeneration. Thus, ideally, experiments would be performed with flies backcrossed to a standard background such as *w¹¹¹⁸*. Did the authors do this, at least for *per⁰*? Since the authors were able to observe rescue of sensitivity, the risk of a background effect explaining the *per⁰* phenotype is reduced. However, they did not report on figure 4A/B the results with *per⁰;uas-per* flies. Since all genotypes presented showed rescue or at least a trend toward improvement, this control should be included.

The *per⁰* flies used in this study have white eyes, as in the paper by Grima et al. (Nature, 2004). We corrected the genotype description in the materials and method (line 492). Therefore, we used white-eye flies as controls in most experiments throughout the study. Additionally, we display the data of CS treated with different concentrations of H₂O₂ (Fig. 2d), showing that PAM neurons in *w¹¹¹⁸* flies are more vulnerable to H₂O₂ than CS. It is thus possible that *white* mutation, in addition to *per* null, contributes to the elevated vulnerability of *per⁰* flies. The data also supports that white-eye flies are better control than red-eye flies for *per⁰*, *w*.

In addition, we now display the data of the PAM neuron counts in the *per⁰; UAS-per* flies in Fig. 4c. No difference in PAM neuron counts was observed between *per⁰; UAS-per* and *per⁰*.

4) The MitoSox experiment in figure 3C does not include a measurement at CT4, the time point at which vulnerability to H₂O₂ was the strongest. The authors might thus have missed an increase in ROS levels at this time point that could underlie the DD rhythm of PAM H₂O₂ sensitivity. This is especially significant as the authors use this result to conclude that “another mechanism [other than ROS rhythms] under the control of circadian clocks gates the vulnerability of PAM neurons in DD.” (line 195-6). This is the only explanation provided for the rather surprisingly different phases of H₂O₂ sensitivity observed between LD and DD in control flies. Also surprising is the very different phase observed under LD in *per⁰*. If light was driving oscillations in LD in control flies, why does *per⁰* show such a different phase compared to wild-type flies?

>The MitoSox experiment in figure 3C does not include a measurement at CT4,

To address this question, we examined ROS levels in PAM neurons DD by monitoring mitoSox levels at CT4, 10, 16, and 22. As presented in the new Fig. S3, no difference was observed between CT4 and other timepoints. Please note that a different microscope was used to capture the images for this set of experiments (since the previous microscope broke down). Hence, we did not combine the data from the new experiment with the data shown in Fig. 3e.

>why does *per⁰* show such a different phase compared to wild-type flies?

It is indeed a difficult question to answer. The output phase of circadian rhythms in physiology and behavior is determined by the combination of zeitgeber and internal clocks. Entrainment phases depend on the characteristics of oscillators, such as period and amplitude, and zeitgeber strength (Bordyugov et al. J R Soc Interface. 2015). It is highly complex how the phase of a rhythm shifts in a given entrainment condition, and thus not surprising that phases of DA neuron sensitivity are several hours different between the wild-type and *per⁰*.

5) The mapping of sensitive DA neurons to a very specific subgroup is really interesting. However, it was done in LD. Is it possible that different neurons might be sensitive under LD and DD, and that this would explain phase differences? It would be worth repeating the experiments in DD for at least MB299B neurons.

To address this question, we performed the short H₂O₂-treatment in DD and monitored MB299B-neurons labeled with GFP at CT4 and CT16. We found a significant loss in MB299B neurons at both timepoints. However, intriguingly, no difference in the magnitude of cell loss was observed between timepoints. Therefore, as you have suggested, it is likely that there is another PAM subpopulation that displays rhythmic vulnerability in DD. The data are shown in Fig. 5c. and described in lines 289-292.

6) Sleep disruptions are emphasized as a symptom of neurodegenerative disorders and Parkinson's disease. It would be worth determining whether sleep consolidation is affected after H₂O₂ treatment.

In fact, we have already shown the analysis of sleep consolidation, by measuring the mean sleep duration per sleep episode (mean sleep episode duration). However, the graphs were not explained clearly, leading to confusion. We re-labeled these plots to avoid misunderstanding. Additionally, to distinguish the "total sleep duration" and "sleep episode duration" more clearly, we now show total sleep duration during 24h, light period (LP), and dark period (DP), replacing the previous data that only displayed sleep duration over 24h. We hope that these modifications make it evident that sleep consolidation is enhanced specifically in DP following H₂O₂ treatment (Fig. 7a-f, lines 346-353)

Minor:

- 1) Line 74, I would use plural for the MBs (bodies instead of body).
- 2) Line 89: Writing that PAM-a1 are "selectively" susceptible might be too strong since two other groups of PAM neuron show some sensitivity.
- 3) Rephrase line 146.

Corrected. Thank you for the suggestions.

4) Line 172-175: Why do flies drink so much more in DD?

We have no idea. Our guess is that water-seeking neural circuit is modulated by light and the loss of light increases the motivation. It has been shown that naïve water-seeking behavior requires some of the PAM DA neurons (Lin et al. Nat Neurosci 2014). It is possible that this circuit is modulated by light and increases DA release under constant darkness, to increase water seeking. But since it is just a speculation, we opt not to add this in the discussion. It is an interesting phenomenon to study in the future.

5) Paragraph starting at line 197: I presume the clkGAL4 driver used here is also expressed in glia and other tissues. I understand that later the authors focus on circadian neurons, but I would suggest mentioning in this paragraph that the driver is not just expressed in clock neurons, and modifying the conclusion on line 204,

We rephrased the statement as suggested (lines 227-239). Furthermore, we included Clk856-GAL4, which has reduced ectopic expression, in the *per* rescue experiment in DD (as explained above, comment (2)). We believe that the use of the second independent Clk-GAL4 driver strengthened our conclusion.

6) Line 245: when introducing split-GAL4 drivers, it would be worth indicating which regions of the MB each innervate.

7) Line 286: Ca²⁺ peaks occur at CT5 and CT17. They would thus be better described as “just before midday and midnight”, rather than “morning and early night”

Corrected as suggested.

8) Line 287: in the sentence “Because Ca²⁺ levels in neuronal cell bodies can be correlated with firing frequency, we wanted to know whether decreased excitability would be neuroprotective” is unclear. This does not seem to flow logically: how does the association between firing frequency and Ca²⁺ levels lead to questioning if excitability was associated with increased vulnerability?

Indeed. We rephrased the statement.

9) Figure 2A/B: Is the “number of PAM neurons,” represented in 2B, cells that co-stain with Red Stinger and GFP, or just Red Stinger-positive cells? Also, the representative images chosen for the per0 brains in 2A do not seem to correlate or represent well the data presented in 2B. In particular, the number of RedStinger-positive cells at Day 1 seems to be much lower than at day 7 or day 14. I must say the image quality is not very good in the file we received, so perhaps some of the staining is obscured by the poor resolution.

Fig. 2b is the number of neurons detected by anti-TH staining, and 2c is by RedStinger.

>Also, the representative images chosen for the per0 brains in 2A do not seem to correlate or represent well the data presented in 2B

We replaced the images in Fig. 2a with more high-resolution images.

10) Figure 4D: although the PDF- 5th sLNv is included in the text as one of the neurons that could project to the PAM neurons, the diagram does not depict it sending projections to the PAM neurons.

We corrected the error, thank you for spotting it (new Fig. 4e)

11) Figure 5B: MB441B should be moved to the rightmost panel (switched with MB315C) to match the order in which these drivers are introduced in the text.

12) Figure 5. It would be preferable to keep the order of GAL4 and split-GAL4 drivers consistent throughout the figure.

Following your advice, we revised the figures (new Fig. 5b, c)

13) Figure 5E: the representative images do not correlate with the quantification in 5F. For the images depicting H₂O₂ exposure at ZT8 and ZT20, it appears that both have about 8-10 neurons left, in contrast with 5F.

We replaced the images to more representative ones (new Fig. 5e).

Reviewer #2 (Remarks to the Author):

The study by Dorcikova et al. investigates whether Parkinson's disease (PD) is associated only with disturbances in sleep and circadian rhythms or whether these disturbances are causally responsible for PD, as they often precede the actual PD symptoms. To answer this question the authors used the well-established model *Drosophila melanogaster*, in which the PD typical loss of dopaminergic (DA) neurons can be induced by oxidative stress (e.g. feeding the flies H₂O₂). The group of Emi Nagoshi has previously shown that H₂O₂ feeding causes the loss of specific DA neurons in the PAM cluster and that this leads to defective climbing behavior suggesting a partial analogy between the PAM

cluster and the mammalian substantia nigra. The PAM neurons comprise ~130 quite heterogeneous neurons that, in addition to locomotion, play critical roles in olfactory associative learning, foraging and sleep.

To test for circadian rhythms in vulnerability to oxidative stress, the authors developed a protocol for short-term H₂O₂ treatment (4h of 5% or 10% H₂O₂) that can be administered at different times of day. They show that this treatment leads reliably to the loss of specific PAM neurons and that the loss depended on the time of day. More importantly, they demonstrate that the rhythm in vulnerability to oxidative stress is controlled by the circadian clock since it persists in the absence of external Zeitgebers and is absent in arrhythmic *per0* mutants. Moreover, they show that arrhythmic mutants (*per0* and *tim0*) possess fewer DA neurons in the PAM cluster even without H₂O₂ treatment and that they lose more DA neurons in response to oxidative stress. Thus, the circadian clock controls the magnitude and temporal sensitivity of the PAM neurons to oxidative stress. Subsequently, the authors performed several critical control experiments that confirmed that it is indeed the circadian clock in specific clock neurons (the PDF-positive LNvs) that regulates the survival of PAM neurons after oxidative stress. Furthermore, they identified the specific PAM neurons (PAM- α 1) that were predominantly killed by oxidative stress and showed that the LN_d clock neurons are presynaptic to them. They demonstrate that the PAM- α 1 neurons show bimodal rhythms in intracellular Ca²⁺, which are absent in *per0* mutants and show that the loss of these neurons affects sleep but not climbing behavior. Finally, the authors investigated the interplay between genetic factors, environmental risk factors and age on the loss of DA neurons in the PAM cluster and found that all factors contribute. This confirms the multi-hit hypothesis for developing PD.

Overall, the study is carefully conducted and clearly shows that disruption of the circadian clock promotes dopaminergic neurodegeneration in flies. The results obtained suggest that mutations in circadian clock genes are an additional risk factor for the development of PD, along with environmental factors and age. These findings could also be relevant for humans and significantly expand our knowledge of the causes of PD.

Nevertheless, the study has several weaknesses that need to be improved before publication. In some places, relevant literature is not cited or discussed and alternative hypotheses are not mentioned. Furthermore, the methods used are not always adequately described.

We thank you for the positive and constructive comments on our manuscript. We thoroughly revised the manuscript following your advice. All the changes introduced in the revised manuscript are highlighted in a marked-up copy of the manuscript.

Specific comments:

Lines 61-72: citing the review of Hall from 2003 and an own review from 2019 in respect of the clock neurons and the clock network is certainly fine, but what about the recent work of Shafer et al. (2022, eLife 11) and Reinhard et al. (2022, JCN 530 and Front Physiol 19) that describe the clock network and their synaptic connections in greater detail? This work needs to be cited.

Agreed and cited these papers (line 68).

Lines 72-74: again, only the authors' own work is cited. What about the work of Liang et al. (2019, Neuron 102) showing that the circadian clock controls ring neurons in the ellipsoid body via PPM neurons? Since PPM neurons are also dopaminergic neurons, this work is very relevant to the present study (see also later).

Agreed and cited the paper (line 75).

Lines 139-143, Figure 2 and Material and Methods: w1118 and CantonS (CS) flies were used as controls for *per0* flies, but none of these stocks is listed in M & M. I suppose that *per0* flies are red

eyed, thus CS flies are probably the better controls. Do you have an explanation for why the w1118 flies are as sensitive to 10% H₂O₂ as the per⁰ mutants (Fig. 2D)?

Thank you for spotting our omission. We now include w¹¹¹⁸ and CS in the Materials and Method (line 498).

> I suppose that per⁰ flies are red eyed,

In fact, per⁰ used in this study have white eyes, as in the paper by Grima et al. (Nature, 2004). As described in the response to comment 3) by Reviewer #1, white mutant strain is a better control genotype in this study.

>Do you have an explanation for why the w1118 flies are as sensitive to 10% H₂O₂ as the per⁰ mutants (Fig. 2D)?

We were surprised to find that w¹¹¹⁸ was more sensitive to H₂O₂ than CS, although to a lesser extent than per⁰. Since white encodes an ATP binding cassette (ABC) transporter, which forms a dimer with Brown or Scarlet to transport precursors of eye pigments. These transporters are also necessary for the synthesis of biogenic amines in neurons, and consequently w mutation is reported to reduce the levels of biogenic amines (Borycz et al. J Exp Biol 2008). Reduced levels of dopamine might contribute to the observed phenotype; however, reduction in dopamine levels is neuroprotective (Bayersdorfer et al. Neurobiol Dis, 2010). Thus, how w modulate dopamine neuron survival is unknown and likely involves non-cell-autonomous mechanisms. It is an interesting question to pursue in future studies.

Figure 2: Are the images in 2A superpositions of several confocal stacks or do they show only one confocal image? How thick are the individual confocal images? The image of the 1-day-old per⁰ fly appears to contain fewer stinger-positive PAM neurons than the images of the older flies, but based on the quantification in Fig. 2C, this is not representative. The many small symbols (data points) in the box plots make it hard to distinguish the white (w1118) and gray (Canton-S) plots. Please try to optimize the pictures. In my opinion, it is enough to indicate the number of analyzed brain hemispheres below the box plots.

In the revised manuscript, we replaced images in the Fig. 2a with high-resolution versions. These are maximum z-projections of part of PAM neurons. Since PAM neurons are numerous and located across approximately 50 z-stacks at 1 μm intervals, if we project all of them into a single image, it will be simply illegible. Therefore, we have to take only about 10 confocal Z-stacks in the middle of the PAM cluster to generate a representative maximum Z-projection image, which captures approximately half of the neurons in the PAM cluster.

We have modified the colors of the Fig.2c box plots to increase the clarity. We have to keep the data points as it is the requirement of the journal.

Lines 199-205: most people use the Clk856-gal4 line for driving expression in all clock neurons. Please explain briefly why you used Clk1982-gal4 for rescuing per in all clock neurons. Are there any differences between the two lines?

Clk1982-GAL4 is highly expressed in most clock neurons, except in the LPN, but has an ectopic expression in non-clock cells, such as the Kenyon cells. In our hands, the effect of reported ectopic expression was negligible (Kolzov et al. PLOS Genet 2020), therefore we originally used only Clk1982-GAL4. However, we acknowledge that Clk856-GAL4 covers the LPN and has reduced ectopic expression, better suited for this study. Therefore, we included Clk856-GAL4 to rescue per⁰ in DD, which we believe strengthened our finding. The new results are displayed in Fig. 4c and described in lines 227-239.

Line 208: the Pdf-gal4 line was not generated by Renn et al. (1999) but by Park et al. (2000, PNAS 97). Renn et al. did also not show that the fifth s-LNv does not express PDF. The fifth s-LNv was first

described as PDF negative by Kaneko et al. (1997, J Neurosci 17), whereas Rieger et al. (2006, J Neurosci 26) first functionally characterized the fifth s-LNV as an evening neuron. Please correct your citation.

Thank you for spotting our errors. Citations were corrected (line 219).

Lines 266-269: What do you mean by visual inspection? Did the hemibrain connectome data reveal any synapses between the LNds and MB299B neurons, and if yes, how many? You can even study Shafer et al. (2022), since these authors might have already reviewed this.

Apologies for the confusing statement. Although chemical synapses between the LNvs or LNds and PAM- α 1 neurons are not annotated in the hemibrain data set, close inspection using the NeuPrint tool revealed close contacts between the axonal projections of the LNds and MB299B neurons. We rephrased the statement in the revised manuscript (lines 295-297). (Shafer et al. (2022) paper does not investigate connectivity between PAM neurons and clock neurons).

Figure 6 and Material and Methods: I did not completely understand how you generated Figure 6C and D. As you stated in Material & Methods Ca²⁺ imaging could be performed in a single fly for up to 6-8 hours and imaging was started every 6 hours to cover the entire day. How many flies could you record at once, and how many cells you imaged on average in each single fly? How similar were the fluorescence intensity values in different PAM neurons within the same fly? How did you detect the death of a fly, and how did you decide when to stop measurements on individual flies and discard the data? If I understood it right, at certain time points, the diagram for *per0* mutants includes the measurements of only 2 flies. At how many time points was n=2? Could the low number of tested flies explain the irregular shape of the curve? In particular, between CT15 and CT18 there is a sudden jump to higher values of fluorescence intensity. Did this jump coincide with a lower number of flies tested? Would there be a possibility to color code the points so that one can immediately tell from how many flies was recorded?

We acknowledge that the GCaMP imaging method description was not detailed. We include more details in the revised manuscript (lines 568-577).

> If I understood it right, at certain time points, the diagram for *per0* mutants includes the measurements of only 2 flies.

You are right in pointing out that the number of *per0* flies could be improved. Therefore, we repeated experiments on both genotypes. The updated data are displayed in the revised Fig. 6c and d. In fact, a peak around CT17 in *per0* is more pronounced with more data points.

The new data are created from an average of 8 *w¹¹¹⁸* flies (5 to 10) and an average of 8 *per0* (5 to 12). This represents an average of 79 neurons per timepoint in *w¹¹¹⁸* (53-140) and an average of 76 neurons per timepoint for *per0* flies (44-131).

>Would there be a possibility to color code the points so that one can immediately tell from how many flies was recorded?

Please see the figures below. In both genotypes, color codes are:

n=5 red; n=6 blue; n=7 orange; n=8 black; n=9 magenta; n=10 green; n=11 grey; n=12 purple.

As we find that these plots do not increase the clarity of the data and can be more confusing, we decided not to display them in the manuscript.

Lines 364-366: what do you mean with “a synaptic structure that indicates fast neurotransmitter signaling was not detected between the LNds and PAM- α 1 neurons in the connectome data”? In the results section, you considered it very likely that these two neurons have synaptic contacts with each other and you showed this even via trans-Tango. Now you say that there are no synapses between the two types of neurons. Please clarify this discrepancy.

Nevertheless, the discussed paracrine, modulatory input from the clock neurons is very likely, since all of them are neuropeptidergic. I wonder why you exclude a neuropeptidergic input to the PAM- α 1 neurons from the s-LNvs via PDF and/or sNPF. The fibers of these two types of neurons are certainly close enough to enable paracrine signaling. At least for PDF this could be easily tested by cAMP-imaging in the PAM- α 1 neurons during the application of PDF. Furthermore, the presence of the PDF-receptor on the PAM- α 1 neurons could be checked using a PDFR-gal4 line. Since you could prevent the loss of PAM neurons by expressing PER only in the PDF neurons (Figure 4), a signalling from the PDF neurons to the PAM- α 1 neurons appears very likely. This is actually the simplest explanation for your findings.

>Please clarify this discrepancy.

Indeed this was our error. This sentence was deleted and

>I wonder why you exclude a neuropeptidergic input to the PAM- α 1 neurons from the s-LNvs via PDF and/or sNPF.

We cannot formally exclude this possibility; however, it is rather unlikely that PDF released from the s-LNvs is received by the PAM- α 1 neurons via PDFR. According to the publicly available single-cell sequencing data (Li et al. Science 2022), only 3% of PAM neurons express *Pdfr* gene. The chances that this 3% overlaps with PAM- α 1 is very low, and even if it happens, it would be less than half of the PAM- α 1 subgroup. In fact, we used *Pdfr-myc* flies (Im and Taghert, J Comp Neurol. 2010) to investigate whether myc signal co-localizes with PAM- α 1 neurons by immunostaining. But the experiment was inconclusive as we did not observe any Myc signal even within clock neurons.

>At least for PDF this could be easily tested by cAMP-imaging in the PAM- α 1 neurons during the application of PDF.

For the reasons described above, we think that this experiment is beyond the scope of this paper.

Lines 372-388: The bimodal Ca^{2+} rhythms in the MB299-labelled neurons are very interesting. Similar bimodal Ca^{2+} rhythms have been found in ring neurons of the ellipsoid body neurons by Liang et al (2019, see above). These ring neurons are rhythmically modulated by DA neurons of the PPM3 cluster, which get neuromodulatory input from the PDF neurons. Thus, this situation strongly

resembles the findings in your study and I do not understand why this is not discussed and the Liang et al (2019) paper is not even cited.

Agreed and discussed the paper in the revised manuscript (lines 424-426).

Lines 408-410: Lyons and Roman (2008 Learn Mem 16) have already shown that the circadian clock modulates short-term memory. Later, Chouhan et al (2015 Curr Biol 25) even showed that flies remember the time of day, an ability that requires the input from the circadian clock to memory centers in the mushroom bodies. Citing these papers may help to discuss the non-motor symptoms of PD.

Agreed, and included the citations as suggested. However, circadian rhythms in memory formation are disputed by Leslie Griffith's lab (Flyer-Adams et al. J. Neuro 2020), therefore, this paper is also cited (lines 466-467).

Figure 8: The sudden death of all *per⁰* mutants treated with H₂O₂ on day 31 is hard to understand. You state that this experiment was repeated twice. It is hard to believe that all flies died exactly at the same day in both experiments. How many flies are included in the survival assay? Have they been kept in just one single large food vial or several food vials? How often food was changed? I did not find any description of this experiment in Material & Methods. This description must be added, and I strongly recommend to repeat this experiment once more.

Following your suggestion, we repeated lifespan experiments (new Fig. 8c, d) and described the methods in detail (lines 618-625). The revised data are from 3 or 4 independent experiments per genotype and treatment, in total of n = 324 *w¹¹¹⁸* flies for control and n = 262 for H₂O₂ treatment in *w¹¹¹⁸*, n = 290 *per⁰* flies for the control and n = 303 for the H₂O₂-treated *per⁰* group. The sudden death of H₂O₂-treated *per⁰* at day 31 was not observed in the 3 new independent experiments. Nevertheless, the H₂O₂ treated *per⁰* flies had a significantly reduced lifespan compared to the control group.

Minor points:

Line 117: Remove the fullstop after "post-treatment".

Lines 145-147: please change into: "...showed a significant loss of PAM neurons as compared to the control group..."

Corrected.

Lines 172-173: Just for interest: do you have an explanation for why flies drink three to five times more H₂O in DD than in LD?

We do not know either. Our guess is that water-seeking neural circuit is modulated by light and the loss of light increases the motivation. It has been shown that naïve water-seeking behavior requires some of the PAM DA neurons (Lin et al. Nat Neurosci 2014). It is possible that this circuit is modulated by light and increases DA release under constant darkness, to increase water seeking.

Reviewer #3 (Remarks to the Author):

Dorcikova et al lay out an interesting and exciting study examining links between circadian rhythm and loss of dopaminergic neurons as a potential mechanism occurring in Parkinson's. The manuscript is very well written and clear with not observable typos or missing details that I can see.

We are pleased to know that you find our work exciting. We made our best effort to address the remaining concern, as summarized below. All the changes introduced in the revised manuscript are highlighted in a marked-up copy of the manuscript.

The study uses flies to initially recapitulate a loss of DA neurons in response to H₂O₂, noting a circadian influence in the effect depending on time of delivery of the insult. They then examine DA neuronal loss in flies mutant for circadian clock components, noting a loss of neurons in the *per0* mutant and a timeless mutant. Given that the chemical insult is delivered by feeding, they then examine any role for starvation or dehydration and find no role.

The key population of neurons are the PAM group and they examine synaptic linkage between circadian neurons and PAM neurons and find connections, data that is bolstered by the *Janelia Farm* connectome data. A subpopulation of the PAM neurons is then genetically defined and these are examined for their resting calcium levels in a 24h period, with a dramatic change in the calcium rhythms altered in the *Per0* mutant. This data is interesting, but more descriptive than mechanistic as it stands. The subpopulation of PAM neurons are associated with the mushroom body, and may regulate sleep. The paper then studies the effects of the H₂O₂ administration on sleep and finds an increase in nighttime sleep. I'm not so comfortable with the specificity of this experiment, as the H₂O₂ delivery is systemic, but the conclusions based on the loss of the subset of DA neurons – so the linkage is limited here.

Please see the response to Reviewer#1, comment 1).

I like this study a great deal and feel that it opens up a productive area of study. My concerns are with the linkage between the feeding and the effects, is ROS in either the DA neurons or the Pdf neurons eliciting these effects. I'm assuming that is the claim, but it would be good to link ROS, either the pdf or DA neurons more directly than the presence of increased MitoSOX detected ROS in the PAM neurons. I would suggest a rescue of the H₂O₂ excess in either the pdf neurons or the PAM neurons using UAS-catalase (or some other anti-oxidant transgene – UAS-Trx, UAS-GST?) to reduce ROS specifically in these neurons to define whether ROS toxicity is causing this effect in pdf neurons, or DA neurons.

This point is well taken. Following your suggestion, we expressed UAS-catalase within PAM neurons and performed the H₂O₂ treatment in LD and DD. Targeted catalase expression effectively prevented H₂O₂-induced PAM neuron loss in all treatments. We believe that these results strengthen the conclusion that the elevation of ROS levels directly triggers PAM neuron loss. These results are included in Fig. 3c and d, and described in lines 186-194.

Overall though, this is a meticulous and novel study and potentially lays out mechanisms for the two-hit theory of insult causing PD

REVIEWER COMMENTS

Reviewer #1 (Remarks to the Author):

The authors have significantly improved their manuscript, responding to most of my concerns. There is however a couple of issues that linger. Both are related to the functional consequences of H₂O₂ treatment.

1) The authors attempted to address my first major comment by inhibiting synaptic transmission in PAM1a neurons. The sleep phenotype observed with Shi-ts partially differs from that observed with H₂O₂ phenotype, which raises questions as to whether the loss of these neurons is really responsible for the H₂O₂ sleep phenotype. The authors propose that perhaps only partial inhibition occurred, or that the varying observations are the result of using different temperatures across experiments. Both are possible of course, but I wonder why the authors did not simply measure sleep after H₂O₂ treatment at 29C to avoid the potential confound of temperature. Also, the authors might have been able to ablate the neurons with HID, or completely suppress them with KIR. Finally, the authors did not attempt to determine whether H₂O₂ and PAM1a inactivation (even if partial) have additive effect or not, as I had suggested. They did not explain why. Thus, whether the sleep defect after H₂O₂ results from the loss of PAM1a neurons remain unclear. At the very least, the authors need to acknowledge that clearly.

2) Reviewer #2 had concerns with the sudden death observed around day 35 after H₂O₂ treatment of per0 flies. The authors added new data to figure 8C-D, but I have a hard time understanding what was done, based on how they responded to the reviewer's comments and the survival curves themselves. Are the curves now shown based solely on the new data, or do they combine old and new data? If the data now shown are all new, then there is still quite a bit of flies dying around day 35. This sudden death would be reproducible, though with variable amplitude. I would then suggest to show both the new and old data (the latter in a supplemental figure). This reproducible death around day 35 would clearly indicate an important impact of per on survival after H₂O₂ treatment. However, if the survival curves now shown are combining old and new data, then there seems to be a significant problem. Indeed, the only difference in the per0 curves would be coming from the death that occurred at day 35 in the first two experiments (rate of death are otherwise quite similar over time). Indeed, comparing the curves now shown to the old ones, I am guessing that the massive day-35 mortality was not observed in the two most recent experiments. This lack of reproducibility would be problematic

Reviewer #2 (Remarks to the Author):

In my opinion, manuscript is carefully revised and I have no further comments.

Reviewer #3 (Remarks to the Author):

I feel that the authors have addressed all the points made by all the reviewers openly and honestly. I'm happy with the result.

Reply to Reviewers' comments

Comments from Reviewers are shown in blue font and our replies are in black.

Reviewer #1 (Remarks to the Author):

The authors have significantly improved their manuscript, responding to most of my concerns. There is however a couple of issues that linger. Both are related to the functional consequences of H₂O₂ treatment.

Thank you for your time and effort to review our manuscript. Below are our answers to the remaining concerns. A marked-up copy of the revised manuscript is uploaded together with a clean copy.

1) The authors attempted to address my first major comment by inhibiting synaptic transmission in PAM1a neurons. The sleep phenotype observed with Shi-ts partially differs from that observed with H₂O₂ phenotype, which raises questions as to whether the loss of these neurons is really responsible for the H₂O₂ sleep phenotype. The authors propose that perhaps only partial inhibition occurred, or that the varying observations are the result of using different temperatures across experiments. Both are possible of course, but I wonder why the authors did not simply measure sleep after H₂O₂ treatment at 29°C to avoid the potential confound of temperature. Also, the authors might have been able to ablate the neurons with HID, or completely suppress them with KIR. Finally, the authors did not attempt to determine whether H₂O₂ and PAM1a inactivation (even if partial) have additive effect or not, as I had suggested. They did not explain why. Thus, whether the sleep defect after H₂O₂ results from the loss of PAM1a neurons remain unclear. At the very least, the authors need to acknowledge that clearly.

We apologize that our explanation was unclear. We initially tried to genetically ablate PAM- α 1 neurons as you have suggested; however, we encountered a number of problems during crosses. It was exceptionally difficult because, to avoid ablating cells during development, we needed to include *tubulin-GAL80^{ts}* to trigger adult-specific PAM- α 1 ablation. We could not manage to establish the right genotype of flies.

Expressing *UAS-Kir2.1* is also a possibility, but to avoid unrelated effects during development, it has to be induced only during adulthood. Please see Fig. S5 as a reference; this was an example with a more broadly expressed R58E02 driver. Based on these data, we judged it's better to avoid expressing Kir2.1 in PAM- α 1 neurons during development. To conditionally silence the PAM- α 1 neurons, it is necessary to add *tubulin-GAL80^{ts}*. This approach also requires a temperature shift to 29-30°C, in addition to having a more complex genotype.

>Both are possible of course, but I wonder why the authors did not simply measure sleep after H₂O₂ treatment at 29°C to avoid the potential confound of temperature.

This idea has escaped our mind, though we are not certain if this could add more clarity to data interpretation. But it is an interesting experiment to perform in the future to more precisely dissect the neural circuit controlling sleep and how neuronal death affect it

>Finally, the authors did not attempt to determine whether H₂O₂ and PAM1a inactivation (even if partial) have additive effect or not, as I had suggested.

We did not do this experiment because the sleep increase is likely reaching the maximum by only the H₂O₂ treatment, especially nighttime sleep. Even if there is an additive effect on sleep increase with H₂O₂ and PAM- α 1 ablation, we may not be able to detect it due to the fact that the sleep amount reaches the upper limit. Therefore, we think that the data obtained from this experiment will be inconclusive.

>Thus, whether the sleep defect after H₂O₂ results from the loss of PAM1a neurons remain unclear. At the very least, the authors need to acknowledge that clearly.

You are absolutely right. We added the statement acknowledging the uncertainty in the revised manuscript (line 366-367).

2) Reviewer #2 had concerns with the sudden death observed around day 35 after H₂O₂ treatment of *per⁰* flies. The authors added new data to figure 8C-D, but I have a hard time understanding what was done, based on how they responded to the reviewer's comments and the survival curves themselves. Are the curves now shown based solely on the new data, or do they combine old and new data? If the data now shown are all new, then there is still quite a bit of flies dying around day 35. This sudden death would be reproducible, though with variable amplitude. I would then suggest to show both the new and old data (the latter in a supplemental figure). This reproducible death around day 35 would clearly indicate an important impact of *per* on survival after H₂O₂ treatment. However, if the survival curves now shown are combining old and new data, then there seems to be a significant problem. Indeed, the only difference in the *per⁰* curves would be coming from the death that occurred at day 35 in the first two experiments (rate of death are otherwise quite similar over time). Indeed, comparing the curves now shown to the old ones, I am guessing that the massive day-35 mortality was not observed in the two most recent experiments. This lack of reproducibility would be problematic

We again apologize for not explaining it clearly. The new data shown in Figure 8c-d are the average of all data, including both the old data set (with sudden death at day 35) and the new data set. Additionally, we have separately analyzed the new data set only. The data are shown below:

As you can see, the water-treated control group and the H₂O₂-treated group have a significant difference in life span (**p<0.001, Log-rank test). You can also appreciate that the shape of the overall survival curves is different from the ones from the previous data (old data set only). In the new data set, there is a more gradual death of the flies at younger ages and also the curve is getting steeper after day 35. However, the main point is that, in both data sets, there is a statistically significant difference in lifespan between the control and the H₂O₂-treated groups. Please also note that we did not claim that the importance was a sudden death around day 35. We believe that removing a data set by a subjective assessment is incorrect, even though the conclusion of the experiment will be unchanged. Now that we have the data of in total of n = 290 *per⁰* flies for the control and n = 303 for the H₂O₂-treated group from 3 to 4 independent experiments, we are confident that these data are reliable.

Reviewer #2 (Remarks to the Author):

In my opinion, manuscript is carefully revised and I have no further comments. Thank you.

Reviewer #3 (Remarks to the Author):

I feel that the authors have addressed all the points made by all the reviewers openly and honestly. I'm happy with the result. Thank you.

REVIEWER COMMENTS

Reviewer #1 (Remarks to the Author):

My first comment was addressed, thank you. For the second, I believe the authors made an error with the figure they presented in their rebuttal. This figure is supposed to show the two most recent *per0* survival experiments, with or without exposure to H₂O₂, to demonstrate that in these experiments there was premature death in *per0* flies exposed to H₂O₂, as observed in the initial experiments. However, the *per0* control curve (in grey) seems to me to be identical to that shown on the revised figure in their manuscript, which combines old and new experiments (figure 8D). I am therefore concerned that in the figure presented in their rebuttal, the authors have mistakenly included all experiments for the control, instead of only the most recent. Since in the first experiments the control died much more slowly, I would guess the actual control curve for the most recent experiments will be closer to the experimental one. Please correct this error to determine if indeed there is reproducible premature death in both the old and new experiments.

Reply to Reviewer's comments

Comments from Reviewer are shown in blue font and our replies are in black.

Reviewer #1 (Remarks to the Author):

My first comment was addressed, thank you. For the second, I believe the authors made an error with the figure they presented in their rebuttal. This figure is supposed to show the two most recent per0 survival experiments, with or without exposure to H₂O₂, to demonstrate that in these experiments there was premature death in per0 flies exposed to H₂O₂, as observed in the initial experiments. However, the per0 control curve (in grey) seems to me to be identical to that shown on the revised figure in their manuscript, which combines old and new experiments (figure 8D). I am therefore concerned that in the figure presented in their rebuttal, the authors have mistakenly included all experiments for the control, instead of only the most recent. Since in the first experiments the control died much more slowly, I would guess the actual control curve for the most recent experiments will be closer to the experimental one. Please correct this error to determine if indeed there is reproducible premature death in both the old and new experiments.

We realize that our explanation was unclear, thank you for pointing it out.

The plot presented in the previous rebuttal letter was generated only from the new data (same as plot C of the figure below). In Fig. 8d of the previous manuscript (plot A), we displayed per0 water-treated control from the new data and per0 H₂O₂-treated data combined with both old and new data. This was because the first experiment was terminated at day 56 when approximately 60% of water-control flies were still alive (plot B), whereas in the new experiment (consisting of three independent replicates), we continued observation until all the flies were dead (plot C). We, therefore, thought that it would make more sense to combine only the data that were collected until all animals have died.

It is possible to combine old and new data sets for both groups, as shown in plot D. However, now that we reflect more on this issue, we think it is most appropriate to display the data that were collected until all animals have died in both groups, i.e. new data only (plot C). For this reason, in the third revision of this manuscript, we replaced Fig.8d with the plot generated only from the new data for both groups (same as plot C).

Figure. Different representations of the lifespan experiments. *** $p < 0.001$ and **** $p < 0.0001$ (Log-rank test).

Above all, please note that, in all plots, there is a statistically significant difference between the control and the H₂O₂ groups. The shapes of the lifespan curves are different between experiments, which underscores that lifespan experiments can be influenced by many subtle environmental factors. However, importantly, the results remain unchanged whether we separately display them or combine them. Thus, we are confident that the results are reproducible. We hope that our explanation is clearer. We again thank you for your keen observation.

REVIEWERS' COMMENTS

Reviewer #1 (Remarks to the Author):

Thanks for the clear explanations. I'm glad to see that the difference between genotypes was indeed reproducible across experiments. The revised figure looks good!